# Inhibition stabilization is a widespread property of cortical networks

**Alessandro Sanzeni[1,2†], Bradley Akitake[1†], Hannah C Goldbach[1], Caitlin E Leedy[1], Nicolas Brunel[2], Mark H Histed[1*]**

[1]NIMH Intramural Program, National Institutes of Health, Bethesda, United States; [2]Department of Neurobiology, Duke University, Durham, United States

**Abstract** Many cortical network models use recurrent coupling strong enough to require inhibition for stabilization. Yet it has been experimentally unclear whether inhibition-stabilized network (ISN) models describe cortical function well across areas and states. Here, we test several ISN predictions, including the counterintuitive (paradoxical) suppression of inhibitory firing in response to optogenetic inhibitory stimulation. We find clear evidence for ISN operation in mouse visual, somatosensory, and motor cortex. Simple two-population ISN models describe the data well and let us quantify coupling strength. Although some models predict a non-ISN to ISN transition with increasingly strong sensory stimuli, we find ISN effects without sensory stimulation and even during light anesthesia. Additionally, average paradoxical effects result only with transgenic, not viral, opsin expression in parvalbumin (PV)-positive neurons; theory and expression data show this is consistent with ISN operation. Taken together, these results show strong coupling and inhibition stabilization are common features of the cortex.

**\*For correspondence:**
mark.histed@nih.gov

†These authors contributed equally to this work

**Competing interests:** The authors declare that no competing interests exist.

## Introduction

Extensive recurrent connectivity between nearby neurons is a ubiquitous feature of the cerebral cortex (*Braitenberg and Schüz, 2013*; *Lefort et al., 2009*; *Binzegger et al., 2004*; *Thomson and Lamy, 2007*), and theoretical work has shown that the strength of recurrent coupling has a major impact on several computational properties of networks of excitatory (E) and inhibitory (I) neurons. Strong excitatory recurrent coupling can increase the speed of network response to external stimuli (*van Vreeswijk and Sompolinsky, 1996*; *van Vreeswijk and Sompolinsky, 1998*), allow a network to sustain persistent activity (*Amit and Brunel, 1997*), and increase the capacity and robustness of memory storage (*Rubin et al., 2017*). Further, strong recurrent coupling can allow networks to amplify specific input patterns, as well as generate complex spatiotemporal activity patterns in response to briefer inputs (*Murphy and Miller, 2009*; *Hennequin et al., 2018*).

Strong excitatory recurrent coupling, however, can lead to unstable dynamics unless stabilized by inhibition. When recurrent connectivity is weak, excitatory cells can show stable firing rates independent of the activity of inhibitory cells. But in networks with strong recurrent connections, excitatory-to-excitatory (E-E) connections amplify responses so that the excitatory network is unstable if the firing rates of inhibitory neurons are kept fixed. Stable excitatory network operation across a range of firing rates can be restored if inhibitory recurrent connections are sufficiently strong, allowing inhibition to track and balance excitation (*Amit and Brunel, 1997*; *Tsodyks et al., 1997*; *van Vreeswijk and Sompolinsky, 1996*; *van Vreeswijk and Sompolinsky, 1998*; *Renart et al., 2010*). Such network models, with strong recurrent connections rendering the excitatory cells self-amplifying and thus unstable, and requiring inhibition for stability, are called inhibition-stabilized networks (ISNs) (*Ozeki et al., 2009*).

Whether cortical networks function in the ISN regime, in which conditions they do so, and which cortical areas may operate as ISNs remain the subject of debate. One key open question has been

whether cortical networks function as ISNs only during high levels of network activity (as when strong sensory stimuli are used to drive sensory cortical regions), or also operate as ISNs for weak input or even during spontaneous activity states. Merely because a network is stabilized by inhibition during one network state does not imply it is inhibition-stabilized for all network states (*Ahmadian et al., 2013*). For example, the fact that the loss of cortical inhibition can lead to epileptic seizures (*Steriade and Contreras, 1998*) may seem at first sight to imply that cortical networks are inhibition-stabilized. However, this observation leaves open the possibility that cortical networks are inhibition-stabilized only in some network states but not all, and that those inhibition-stabilized states are the ones that generate seizure activity.

One area, cat primary visual cortex (V1), shows behavior clearly consistent with the ISN regime (*Ozeki et al., 2009*). This was established in the presence of sensory stimuli, and thus could not determine whether cat V1 operates as an ISN in the absence of visual stimuli when network activity is lower (i.e. at rest, during spontaneous activity). Based on these data and others, Miller and co-authors later developed a model called the stabilized supralinear network (SSN) (*Ahmadian et al., 2013*; *Rubin et al., 2015*), which explains cat and ferret V1 responses to visual sensory stimuli of different sizes and intensities. The SSN shows ISN-regime operation for strong visual stimuli, but predicts that as sensory stimuli decrease in strength, the network should transition into a non-ISN state.

To examine whether cortical networks operate as ISNs or non-ISNs in different activity levels, optogenetic stimulation can be used to test predictions of ISN models. A strong prediction of inhibition stabilization is *paradoxical inhibitory suppression* — a counterintuitive decrease of inhibitory firing rate as inhibitory cells receive direct input (*Figure 1A–B*, *Figure 1—figure supplement 1*; *Tsodyks et al., 1997*). The paradoxical effect is a result of the strong synaptic coupling between excitatory and inhibitory cells. In an ISN, due to the strong recurrent coupling, more recurrent excitation is withdrawn after inhibitory stimulation than the stimulation adds, leading to a net decrease in input to stimulated inhibitory cells (*Figure 1—figure supplement 1*). This decrease in recurrent excitation is the cause of the paradoxical effect. Thus, examining whether inhibitory cells show paradoxical suppression when stimulated should be able to reveal whether cortical areas are operating in the ISN regime.

However, optogenetic ISN studies, largely performed in the mouse, have produced conflicting data about ISN operation during spontaneous activity states. While some paradoxical changes in

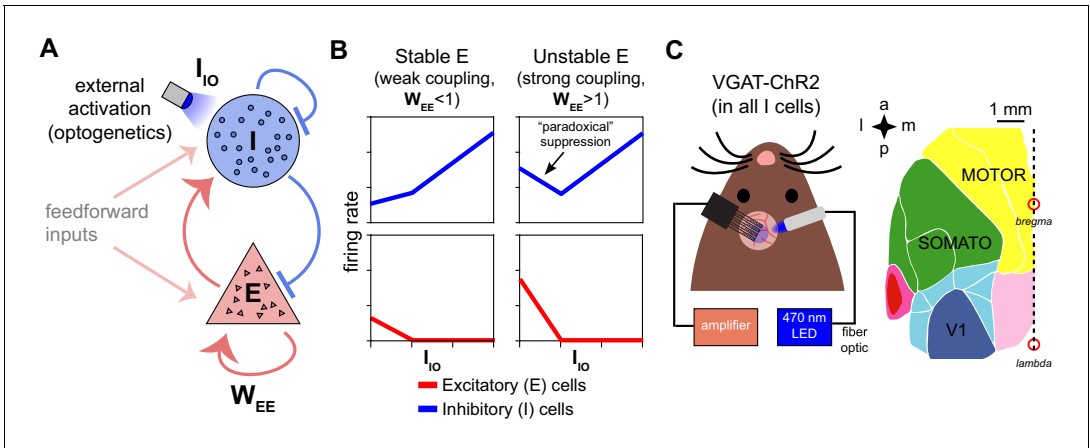

**Figure 1.** Model predictions of excitatory and inhibitory responses to inhibitory stimulation. (**A**) Schematic of model, showing connections between excitatory (E) and inhibitory (I) neuron populations. ($W_{EE}$, a measure of the strength of E-E connections, is the key parameter controlling non-ISN vs. ISN operation, see model discussion in Results.) (**B**) Predictions for average neural responses with weak recurrent coupling (left) and strong coupling (right), when inhibitory cells are externally stimulated ($I_{IO}$). (**C**) Schematic of experiment. Extracellular recordings made in visual (V1), primary somatosensory (SOMATO), and motor/premotor cortices (see *Figure 1—figure supplement 2* for electrode locations; a: anterior, p: posterior, m: medial, l: lateral) while optogenetically stimulating inhibitory cells at the recording site in awake VGAT-ChR2 animals.

The online version of this article includes the following figure supplement(s) for figure 1:

**Figure supplement 1.** Nullclines characterizing the network response at different levels of inhibitory drive.
**Figure supplement 2.** Locations of recording sites.

inhibitory currents have been observed in mouse primary auditory cortex (A1) (*Kato et al., 2017*; *Moore et al., 2018*), several studies have found, in contrast, non-paradoxical effects in inhibitory firing rate or intracellular currents in mouse V1, A1, and primary somatosensory cortex (S1) (*Atallah et al., 2012*; *Moore et al., 2018*; *Gutnisky et al., 2017*). One study that observed paradoxical changes in firing in mouse A1 via stimulation with ArchT (*Moore et al., 2018*) found that those paradoxical effects were not produced via the ISN mechanism, but instead mediated by a circuit where L4 inhibitory cells suppressed L2/3 inhibitory cells (i.e. using feedforward inhibition). Possible explanations for these varying observations include differences between anesthetized and awake states, differences between measurements of firing and intracellular currents, or, as we show experimentally below, differences in which, or how many, inhibitory cells are stimulated (*Litwin-Kumar et al., 2016*; *Sadeh et al., 2017*; *Gutnisky et al., 2017*). Additionally, in mouse V1, one study (*Adesnik, 2017*) found that excitatory and inhibitory currents scale as predicted in the SSN with increasing visual stimulation. But this study of SSN phenomena also leaves open the question whether mouse V1 operates as an ISN without sensory stimuli (see also *Litwin-Kumar et al., 2016*, their Discussion).

Here, we examine whether cortical networks operate as ISNs at rest, by combining optogenetic stimulation of inhibitory cells, extracellular recordings in awake animals from several cortical areas, and theoretical analyses. We find clear evidence for ISN operation even without sensory stimulation in multiple mouse cortical areas and both in superficial and deep layers. By providing multiple experimental and theoretical lines of evidence for ISN operation, our results argue against non-ISN explanations (e.g. against paradoxical suppression arising from one inhibitory population inhibiting another).

We also address whether stimulating a single genetic subclass of inhibitory neurons in V1, the parvalbumin-positive (PV) cells, can produce paradoxical inhibitory effects. We find differences in paradoxical responses between viral and transgenic expression strategies. These differences are explained by an ISN model where the fraction of stimulated PV cells is varied (*Gutnisky et al., 2017*; *Sadeh et al., 2017*), and are supported by histological measurements, supporting the idea that cortical responses to inhibitory input can be highly dependent on the number of stimulated cells (*Sadeh et al., 2017*). Additionally, while synaptic plasticity (*Varela et al., 1999*) such as depression at excitatory synapses can impact network stability, we show via theoretical work (Appendix 1) that the paradoxical effects we observe imply stabilization by inhibition, even in the presence of synaptic plasticity.

Together, these results support the idea that the cortex operates in the ISN regime across a range of areas and brain states.

## Results

### Mouse primary visual cortex is inhibition stabilized

We first describe experiments where all inhibitory neurons were optogenetically stimulated. We expressed an excitatory opsin (ChannelRhodopsin-2; ChR2) in all inhibitory neurons using a transgenic mouse line (VGAT-ChR2). We delivered blue light to the surface of the cortex, while recording activity extracellularly with multi-site silicon recording arrays (*Figure 1*, *Figure 1—figure supplement 2*). Mice were given drops of reward approximately once per minute (Materials and methods) to keep them awake and alert. Because neurons in the superficial cortical layers showed the largest responses to the stimulation light (*Figure 2—figure supplement 1*), we restricted our analyses to units within ≈400 µm of the cortical surface (Materials and methods), which primarily includes neurons in layer 2/3 and upper layer 4 (*Harris et al., 2018*). We sorted units into single- (separated from noise and other units) and multi-units (lower SNR or containing several apparent units) by waveform (Materials and methods; using a more restrictive threshold for unit inclusion does not affect our conclusions, *Figure 2—figure supplement 2*), and we excluded the multi-units from further analyses.

The majority of recorded units were suppressed by light (146/167 single units, 87%; *Figure 2B–D*; 4 animals, 7 recording days). Given that approximately 80% of cortical neurons are excitatory (*DeFelipe et al., 2013*; *Tremblay et al., 2016*), and that extracellular recordings can show several biases in sampling cortical neurons (*Margrie et al., 2002*; *Olshausen and Field, 2005*), this measured percentage could be consistent with either the presence or absence of paradoxical

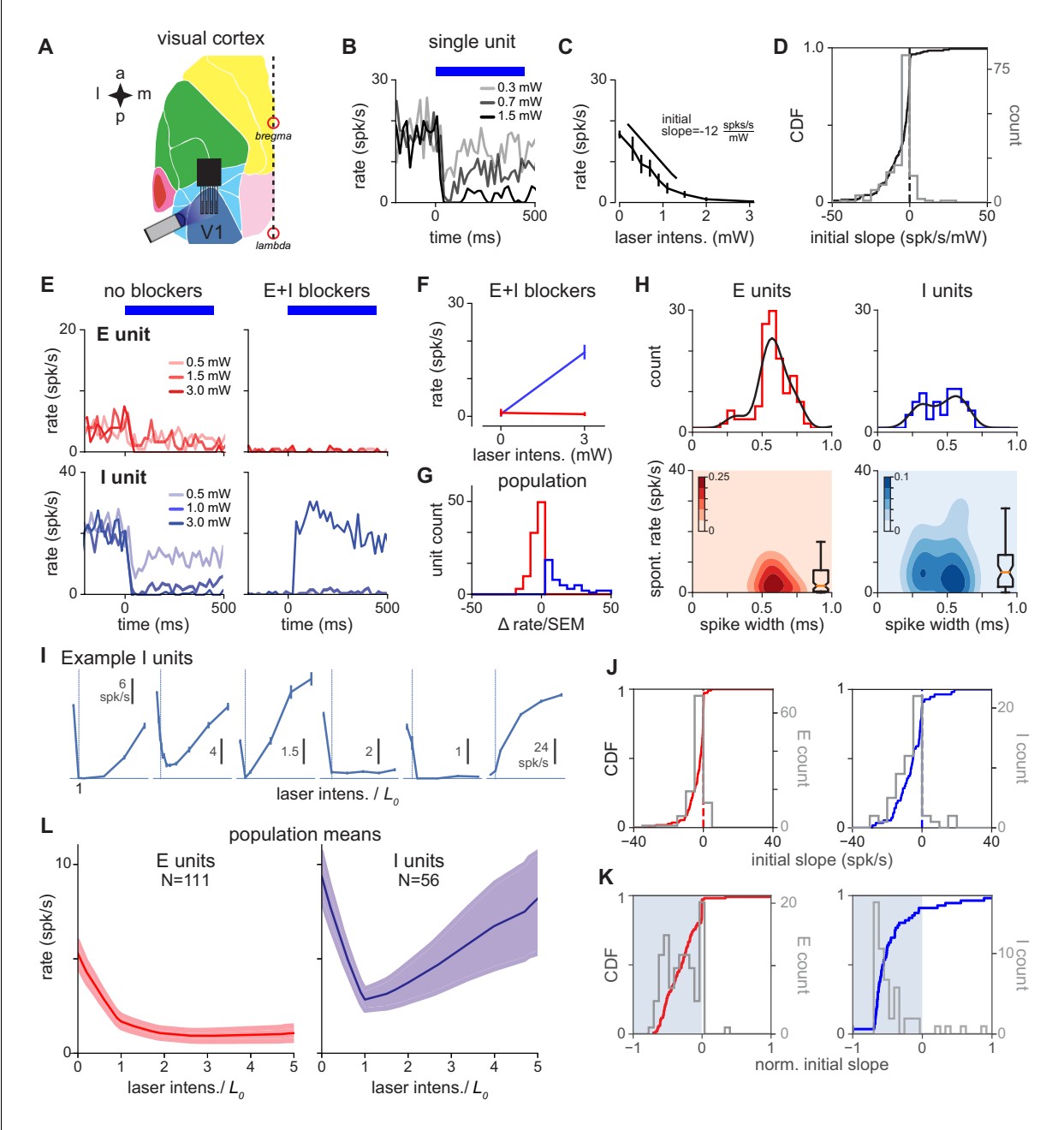

**Figure 2.** V1 response is consistent with inhibition stabilization. (**A**) Schematic of experiment, showing recordings/stimulation in V1. a: anterior, p: posterior, l: lateral, m: medial. V1 recording locations, and intrinsic signal imaging to define V1, are shown in *Figure 1—figure supplement 2*. (**B**) Example recorded unit; greater laser power produces greater firing rate suppression. Blue bar: duration of constant-power laser pulse. (**C**) Average firing rate of example recorded unit (rate is steady-state firing rate, over last 300 ms of laser stimulus). Suppression is quantified via initial slope of firing rate vs laser intensity (Materials and methods). (**D**) Distribution of initial slopes for all recorded units. (**E**) Example cell responses without (left) and with (right) blockers. Top: Cell classified as excitatory; does not respond to laser in presence of blockers (CNQX, APV, bicuculline; Materials and methods). Bottom: Cell classified as inhibitory; increases firing rate to laser in presence of blockers. (**F**) Firing rates for units in (**E**) in presence of blockers; inhibitory cell (blue) shows a large positive change. (**G**) Distribution of firing rate changes to laser stimulus with blockers. Red: E-classified units; blue: I-classified units. (**H**) Spontaneous firing rates and waveform width (time from waveform first local minimum to first local maximum) for units classified as E and I via the method in panel G. (**I**) Example inhibitory units. x-axis: normalized laser intensity. Normalization: to account for variations in tissue optical properties across recording days, we combined data by fitting a piecewise-linear function to the average inhibitory response for each day to find the minimum for that day ($L_0$: always between 0.3 and 2.7 mW, see *Figure 2—figure supplement 6*) and rescaling the laser intensity so $L_0 = 1$ (vertical lines). Most units (91%) show initial suppression; see J,K for initial slopes for all units. (**J**) Distribution of initial slopes for units classified as E (red, left)

*Figure 2 continued on next page*

*Figure 2 continued*

and I (blue, right). (**K**) Distribution of normalized initial slopes ($\frac{\text{init.slope}}{\text{baselinerate}+0.1\text{spk/s}}$), to more clearly show small slopes, for units classified as E and I. Several excitatory cells with very low firing rates in the no-blocker case have near-zero normalized slopes. (**L**) Average population response for units classified as E and I. Inhibitory cells show a prominent paradoxical effect. Errorbars: ±1 SEM (throughout figures, unless otherwise noted).

The online version of this article includes the following figure supplement(s) for figure 2:

**Figure supplement 1.** Unit response to light as a function of depth.

**Figure supplement 2.** Experimental results are robust to quality of unit isolation.

**Figure supplement 3.** Paradoxical effects in V1 are preserved when inhibitory units are classified via pharmacology, via waveform width, and via response at high laser power.

**Figure supplement 4.** Paradoxical V1 inhibitory response does not depend on statistical E/I classification threshold.

**Figure supplement 5.** Waveform drift has no qualitative effect on population response.

**Figure supplement 6.** Reversal point of V1 inhibitory population is typically at a low laser intensity.

**Figure supplement 7.** V1 inhibitory cell responses.

**Figure supplement 8.** Responses of broad- and narrow-spiking V1 (VGAT-ChR2) inhibitory units.

suppression in inhibitory neurons. Therefore, we classified recorded units as inhibitory or excitatory using in vivo pharmacology. (Our results are confirmed by other classification methods: identifying inhibitory neurons using their response at high laser intensity or by waveform width give similar results, see below and *Figure 2—figure supplement 3*).

We first recorded neurons' responses to optogenetic stimulation, and then applied blockers of excitatory and inhibitory synapses (CNQX, AP5, bicuculline; which block AMPA, kainate, NMDA, and GABA-A synapses; Methods). When recurrent synapses are substantially suppressed, inhibitory cells expressing ChR2 should increase their firing rate to optogenetic input, while excitatory cells' firing should not increase. Indeed, by measuring responses to inhibitory stimulation in the presence of E and I blockers, we were able to classify units into two groups (*Figure 2E*). Units in the first class (putative excitatory) were silent for strong stimulation and units in the second class (putative inhibitory) increased their firing rate in response to stimulation. To quantitatively classify cells, we measured the change in activity induced by light when the blockers had been applied (*Figure 2F–G*); cells were labeled inhibitory (33%; 56/167 units) if the change compared with baseline produced at maximum laser intensity was positive (according to Welch's t-test, $p<0.01$), and excitatory otherwise. Choosing different classification thresholds did not affect our results (*Figure 2—figure supplement 4*), examining a subset of cells with the most stable waveforms over time did not affect our results (*Figure 2—figure supplement 5*), and, finally, examining a subset of inhibitory units where waveforms had high signal-to-noise ratio also did not qualitatively change the results (*Figure 2—figure supplement 2*). Therefore, we used these response differences in the presence of blockers to identify recorded inhibitory units.

Pharmacologically identified inhibitory and excitatory units showed differences in waveform width and spontaneous firing rate (*Figure 2H*), although these two factors were insufficient alone to completely classify inhibitory cells (i.e. insufficient to predict whether units would increase firing rate in the presence of blockers). Identified excitatory cells had a wider waveform (0.58 ± 0.01 ms) and lower firing rate (6.0 ± 0.8 spk/s) than inhibitory cells (width 0.46 ± 0.02 ms, rate 10.7 ± 1.7 spk/s). However, inhibitory neurons' widths were broadly distributed, supporting the idea that that some inhibitory classes can have broader waveforms than others (*McCormick et al., 1985*; *Neske et al., 2015*).

Once recorded units were classified as excitatory or inhibitory, we could ask whether the same units before adding blockers (i.e. with the animal in the awake state, with recurrent connections intact) showed paradoxical inhibitory effects. Indeed, inhibitory neurons showed strong suppression when stimulated (*Figure 2I–J*, *Figure 2—figure supplement 7*). The mean response over all recorded inhibitory units was suppressed (*Figure 2L*), and also, the large majority of inhibitory units were suppressed (distributions of response slopes shown in *Figure 2J–K*; responses for all inhibitory cells as power is varied shown in *Figure 2—figure supplement 7*). This paradoxical inhibitory suppression is one piece of evidence that an ISN model is a good description of the upper layers of primary visual cortex. (Below we show that response dynamics and responses at high laser intensity provide additional evidence for an ISN.)

These experiments were done without sensory stimulation, as animals viewed an unchanging neutral gray screen. Because visual cortical neurons' firing rates increase with increasing contrast across a range of overall luminance levels, we expected that we would see little difference between data collected with a neutral gray screen and data collected in the dark. Confirming that expectation, in one experimental session we measured inhibitory responses to stimulation while animals were in the dark (with the blue optogenetic light shielded with light-blocking materials), and found paradoxical suppression (N = 6/6 inhibitory units with initial negative slope, median initial slope $-6.9 \pm 3.5$ spk/s; p = 0.046, median < 0, Mann-Whitney U test).

In an ISN, paradoxical inhibitory suppression arises because exciting inhibitory cells in turn suppresses excitatory cells, which withdraw excitation from the stimulated inhibitory cells. Then, because more excitatory input from recurrent sources is withdrawn from inhibitory cells than excitatory input from optogenetic stimulation is added, the net steady-state response of inhibitory cells is suppression. For this mechanism to hold, inhibitory cells must transiently increase their firing, even if only a small number of inhibitory spikes are generated, before the paradoxical suppression (*Tsodyks et al., 1997*; *Ozeki et al., 2009*). Therefore, we looked for transient increases in inhibitory firing with short latencies after stimulation, and found a small increase in inhibitory firing rate immediately after stimulation, before the larger paradoxical suppression (*Figure 3*). The transient was brief (FWHM = 7.1 ms), and so the actual number of extra spikes fired in the transient was small: only 0.03 extra spikes were fired on average per inhibitory-classified unit per stimulation pulse, although because we used 100 pulse repetitions, the majority of classified inhibitory units (63%, N = 35 of 56), showed a detectable initial transient (*Figure 3—figure supplement 1*).

Another ISN prediction is that inhibitory cells should increase their firing rates at high laser intensities. In a recurrent network operating as an ISN, stimulating inhibitory cells first decreases inhibitory firing rates, but stronger stimulation eventually suppresses excitatory cells until they reach a firing rate where the excitatory network is stable without reactive inhibition. Then, increasing stimulation of inhibitory cells beyond this point produces increases in inhibitory firing rates, as the network moves into a non-ISN regime (*Figure 1C*). Indeed, we observed this in our excitatory and inhibitory identified populations (*Figure 2H*).

The link between paradoxical responses and inhibition stabilization was initially derived (*Tsodyks et al., 1997*) assuming recurrent connections do not change their strength. Short-term synaptic plasticity (*Tsodyks and Markram, 1997*; *Markram et al., 1998*) could modify this link, as synaptic plasticity can influence network stability, for example by reducing the self-amplification of the excitatory network as excitatory rates rise. We thus set out to determine analytically how synaptic plasticity might affect both responses and dynamics in a network model of excitatory and inhibitory neurons (Appendix 1). We find that, even when short-term synaptic plasticity is included in the model, paradoxical responses imply inhibitory stabilization. Therefore, even when our data are interpreted in a modeling framework that includes short-term plasticity, our experimental observations provide evidence for inhibition stabilization.

While our extracellular recording approach has advantages, such as allowing precise measurements of spike rate dynamics, and recording a large number of E and I neurons in the same experiment, one concern might be that misclassification of excitatory waveforms into inhibitory units could mask inhibitory cell responses. However, confirming that our recording procedures accurately measured inhibitory waveforms, (1) paradoxical suppression is maintained when we make our classification thresholds more stringent (restricting analysis to units with highest waveform signal-to-noise ratio, *Figure 2—figure supplement 2*; varying classification threshold in the presence of blockers; *Figure 2—figure supplement 4*), (2) units classified by spike width, or by response at high laser power, also showed clear paradoxical effects (*Figure 2—figure supplement 3*), and (3) inhibitory, but not excitatory, response dynamics showed an initial increase followed by suppression (*Figure 3*, *Figure 3—figure supplement 1*), the response patterns expected for E and I neurons in an ISN. Also, as shown below, in parvalbumin-positive neuron stimulation with viral transfection, using the same waveform-sorting procedures, we observed no average paradoxical effects. Together, these observervations argue that paradoxical effects in VGAT-ChR2 animals do not arise due to waveform sorting, but instead due to the properties of the cortical inhibitory network.

Taken together, these results show that mouse V1 responses are consistent with those of a strongly-coupled network whose activity is stabilized by inhibition.

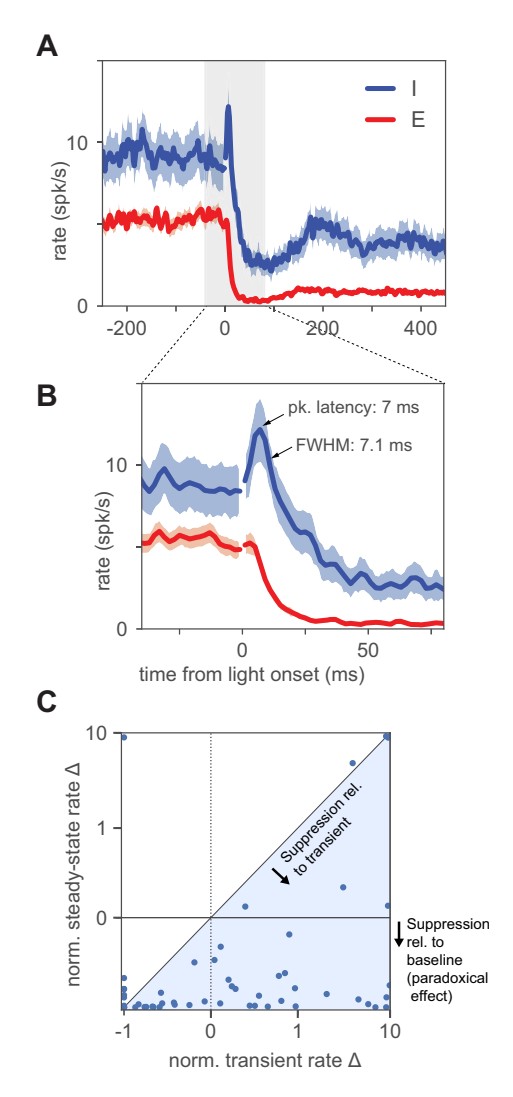

**Figure 3.** Inhibitory neurons show a small increase in firing before paradoxical suppression. (**A**) Average timecourse of neural responses from units classified as excitatory (N = 111) and inhibitory (N = 56). Light pulse is constant-intensity and lasts for 800 ms (high-intensity pulse, 2.6 $\cdot L_0$; stimulation strength larger than $L_0$ is predicted by ISN models to produce larger transients than at $L_0$, though steady-state I rates are larger than the minimum; see also *Figure 3—figure supplement 1*). Data from VGAT-ChR2 animals. Heavy lines: population mean rate, smoothed with a LOWESS filter. Shaded regions: SEM (Materials and methods). The onset of the light pulse produces an artifact in the 0 ms time bin; we drop spikes at that time point (broken lines, time = 0) to remove the artifact. Inhibitory neurons show a brief increase in firing before suppression, and the increase above baseline is statistically significant (inhibitory firing rates over 12 ms window after pulse are greater than a matched-duration interval before pulse, *p*<0.01, KS test). (**B**) Same data as (**A**), enlarged to show the initial transient

*Figure 3 continued on next page*

## Model-based inference of network coupling strength and stability

We next used these data to infer network connection parameters in a standard two-population model that describes the dynamics of population-averaged firing rates of excitatory and inhibitory neurons (see *Equation 6*; Materials and methods). This parameter inference is possible because we obtained E and I response measurements in three pharmacological conditions (no blockers; with E blockers; with E and I blockers; *Figure 4A*), and with data from these three conditions, there are more experimental observations than model parameters (Materials and methods: Model degrees of freedom). The model uses a rectified-linear single-neuron transfer function; using nonlinear transfer functions give similar results (*Figure 4—figure supplement 1* and Appendix 2).

First, we find that the model, despite being overconstrained by the data, is a good description of V1 responses (*Figure 4A*). The model also makes a number of predictions that are verified in the data. It predicts that the inhibitory firing rate slope for high laser intensities should be the same in the no-blocker and E-blocker conditions, as seen in the data (*Figure 4A*, blue: left vs. middle columns; see also *Appendix 2—figure 1*). The model makes three other predictions: that the reversal point of inhibitory cells ($L_0$, the point at which initial negative inhibitory slopes become positive), in all three experimental conditions, should match the point where excitatory cells change their slope to become nearly silent. These effects are also seen in the data (e.g. clearly visible in the no-blocker case, *Figure 4A*, left column). Note that, although these predictions have been derived in a threshold-linear model (Materials and methods), they are general features of network models which show a transition from non-ISN to ISN as the activity level of the excitatory population increases.

One prediction of balanced-state models with strong recurrent coupling (*van Vreeswijk and Sompolinsky, 1996*) is that the total excitatory and inhibitory currents should each be large, although they cancel to produce a small net input. To check this, we computed from the model the currents flowing into excitatory cells (*Figure 4B*). The excitatory and inhibitory currents are each approximately ten times larger than threshold (medians are 10.9 and −8.63, respectively), and their sum is small compared to the magnitudes of E and I currents (median net current/threshold = 2.2). The model also allows

*Figure 3 continued*

(time range here is indicated in (**A**) by gray shaded region). Inhibitory initial positive transient has peak amplitude 3.6 spk/s above baseline, latency to peak 7 ms, full width at half maximum 7.1 ms, and inhibitory rate crosses baseline into suppression at 13.1 ms. (**C**) Size of transient vs. size of steady-state suppression for all I units. The majority of units show steady-state paradoxical suppression relative to both baseline rate and to transient rate. y-axis: steady-state rate change (normalized to baseline; $\frac{rate-baseline}{baseline}$). x-axis: normalized transient; measured from 1 to 12 ms after light onset. Units below the horizontal line at zero show paradoxical suppression; units in the blue shaded region are those whose normalized steady-state rate is lower than their normalized transient rate. A few units show a positive transient and elevated steady-state rate (upper right quadrant); these are likely inhibitory units showing non-paradoxical steady-state rate increases. The online version of this article includes the following figure supplement(s) for figure 3:

**Figure supplement 1.** Many inhibitory cells show an initial transient and also steady-state paradoxical suppression.

us to infer the value of the key parameter for inhibition stabilization, $W_{EE}$ (the excitatory self-amplification). The excitatory network is unstable, requiring inhibition for stabilization, and yielding an ISN, when $W_{EE}>1$. As expected when paradoxical effects are present, the fitted value is greater than 1 (*Figure 4C*: median 4.7, mode 2.5). This value is not much larger than unity, suggesting recurrent coupling is not as strong as some balanced-state models might predict (*van Vreeswijk and Sompolinsky, 1996*; *van Vreeswijk and Sompolinsky, 1998*; *Amit and Brunel, 1997*) but is more consistent with a moderately-coupled network (*Ahmadian et al., 2013*; *Rubin et al., 2015*; *Hennequin et al., 2018*). In Appendix 2, we extend this model analysis to examine networks of spiking neurons which feature a fixed level of input noise, and find qualitatively similar results. The model shows that measured activity is consistent with the mean input to cells being below threshold. This implies the recorded cells operate in the fluctuation-driven regime, matching expectations from cortical response variability (*Shadlen and Newsome, 1994*; *Shadlen and Newsome, 1998*; *van Vreeswijk and Sompolinsky, 1996*; *van Vreeswijk and Sompolinsky, 1998*; *Amit and Brunel, 1997*). Moreover, when input fluctuations are taken into account in the model, mean excitatory and inhibitory currents are found to be of order of the distance between rest and threshold, in agreement with recent estimates suggesting the cortex operates in the loosely balanced regime (*Ahmadian and Miller, 2019*).

Finally, we computed the parameter ranges over which the network is stable, as a function of the time constants of E and I firing rate dynamics, $\tau_E$ and $\tau_I$. We find, by computing the eigenvalues of the connection matrix (Materials and methods) that the network is stable for $\tau_I/\tau_E$ in the interval [0, 5.3], and the observed dynamics are consistent with $\tau_I/\tau_E = 4.4$ and $\tau_E = 7.8$ ms (*Figure 4E*); that is, with excitatory and inhibitory synaptic time constants of the same order. The values of $\tau_{(E,I)}$ (*Figure 4D*) inferred from the response dynamics are within the stable region, providing further support that the ISN model is a good description of the effects we measure. (Note that the time constants of the opsin may increase these estimates, so the values of $\tau_E$ and $\tau_I$, while small, are only upper bounds on the true dynamic response of the network to an instantaneous conductance change.) The ratio between E and I time constants is close to the point at which the network becomes unstable through an oscillatory instability (*Figure 4E*). The fact that the inferred network model describes the data and is stable with $W_{EE}>1$ supports the experimental observation that the underlying cortical network is inhibition-stabilized.

## Inhibition stabilization is present in other cortical areas

The data above show that the superficial layers of mouse V1 are inhibition stabilized. Another open question is how general the ISN regime is across cortical areas. We performed experiments to look for signatures of inhibitory stabilization in two other cortical areas (*Figure 5*): somatosensory (body-related primary somatosensory, largely medial to barrel cortex) and motor/premotor cortex. (See *Figure 1—figure supplement 2* for locations of all recording sites.) As in V1, we recorded units extracellularly, restricted analysis to the superficial layers of the cortex, and examined responses to optogenetic stimulation of all inhibitory cells (as above, using the VGAT-ChR2 line).

In these experiments, we classified neurons as inhibitory based on their activity at high laser intensity. In our V1 experiments, paradoxical effects are clear whether neurons are classified in this way, or with the pharmacological method (*Figure 2—figure supplement 3*). This high-laser-power classification is likely more stringent (i.e. may reject some inhibitory cells if they are suppressed by other

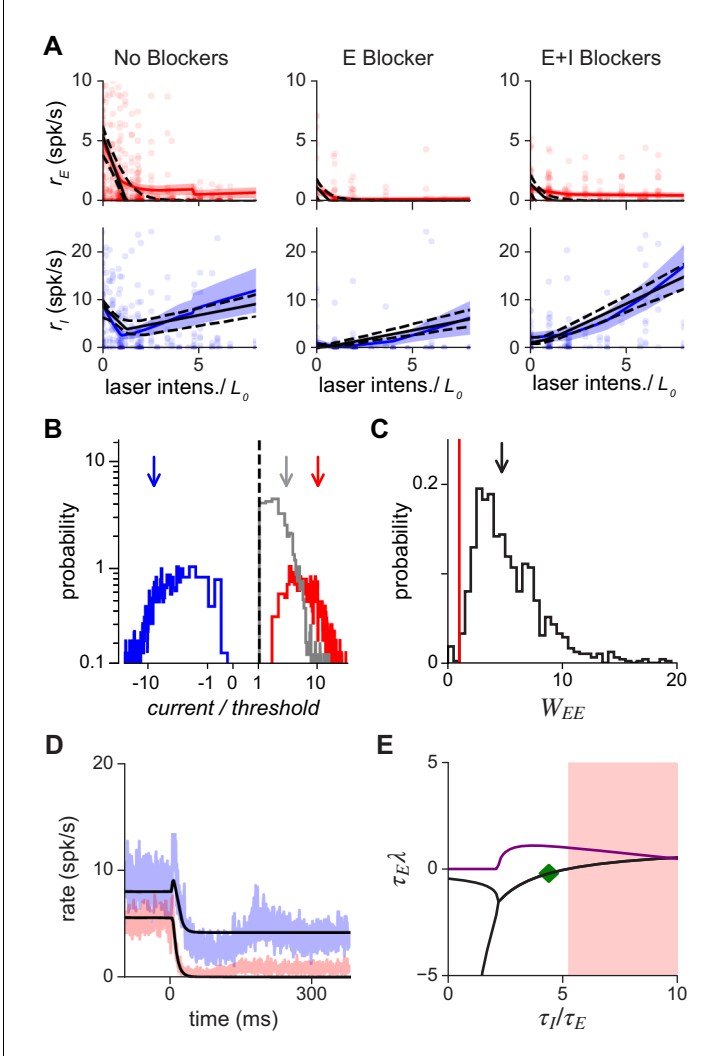

**Figure 4.** Population response is consistent with a network with moderately strong coupling. (A) Average excitatory (red) and inhibitory (blue) response measured in three conditions: without synaptic blockers, with only the excitatory blockers present, and with both E and I blockers present. Model (*Equation 6*) that best fits data: black continuous line; dashed: 1 s.d. obtained by bootstrap; solid red/blue lines: E/I data means; shaded region: ±1 SEM. Because we applied blockers sequentially (generating separate E and E+I blocker measurements) the number of independent observations were increased, allowing the inference of all model parameters (Materials and methods; *Figure 4—figure supplement 2*). Small steps in left panels (no blockers) arise because a subset of experimental days used a maximum laser intensity of $5 \cdot L_0$. (B) Excitatory (red), inhibitory (blue), and net (gray) current influx into excitatory cells predicted by the model. Arrows: medians (over bootstrap repetitions); E: 10.9, I: −8.63; modes: E: 2.5, I: −2.7. (C) Distribution of $W_{EE}$ values compatible with the data; the red line represents the transition point between the ISN ($W_{EE}>1$) and the non-ISN ($W_{EE}<1$) regime. Median (arrow) 4.7; mode 2.5. (D) Estimation of time constants of E and I populations. Black line shows the dynamics resulting from fitting the data (blue: I population; red: E population; shaded region ±1 SEM) with the model for the same laser power shown in *Figure 3*. Best-fit values are $\tau_I = 34.3$ ms and $\tau_E = 7.8$ ms; note that network response dynamics shows faster time constants due to recurrent network effects. The full model provides a good approximation to the dynamics even though it is constrained to simultaneously fit the time constants and the responses at different intensities. (E) Stability analysis. Real (black) and imaginary (purple) parts of the eigenvalues of the Jacobian matrix as a function of the ratio $\tau_I/\tau_E$. Imaginary part (purple) greater than zero signifies the network can show damped oscillations when being driven to a new stationary point. Note that these damped oscillations are not seen in (D) because of rectification. When the real part (black) is greater than zero, the network is unstable (shaded red area). The online version of this article includes the following figure supplement(s) for figure 4:

**Figure supplement 1.** Comparison of linear and nonlinear rate models, and global optimization method.

*Figure 4 continued on next page*

*Figure 4 continued*

**Figure supplement 2.** V1 model parameter stability shown via data bootstrap.

inhibitory cells, due e.g. to heterogeneity of recurrent connections) than pharmacological classification. Supporting that this classification may be more stringent, some pharmacologically-classified inhibitory cells show paradoxical effects but little increase at high power (*Figure 2—figure supplement 8*).

In both the recorded areas (*Figure 5*), inhibitory neurons showed paradoxical effects, supporting that these cortical areas do also operate as an ISN. Similar to the V1 data, and as expected in an ISN, both areas showed a transition to a non-paradoxical response at large laser intensity. Compared to the somatosensory data, the motor/premotor recordings show a smaller average suppression (*Figure 5B*) and more inhibitory units with non-paradoxical effects (rate increases with stimulation; *Figure 5D*). However, in both areas, means and medians are significantly negative (see legend for statistical tests). Thus, the superficial layers of both sensory cortical areas we examined (visual and somatosensory), and one non-sensory area (motor/premotor) showed responses consistent with strong coupling and ISN operation.

## Differences in paradoxical suppression with viral or transgenic opsin expression can be explained by different numbers of stimulated cells

Up to this point, we have studied paradoxical suppression by stimulating an opsin expressed in all inhibitory cells via the VGAT-ChR2 mouse line. A remaining question is whether stimulation of a subclass of inhibitory neurons also yields paradoxical suppression. Even in an ISN, stimulation of any single subclass of inhibitory cells need not produce paradoxical suppression (see Materials and methods and *Rubin et al., 2015*; *Litwin-Kumar et al., 2016*; *Sadeh et al., 2017*; *Gutnisky et al., 2017*).

To study this, we stimulated parvalbumin-positive (PV) neurons, which provide strong inhibitory input to other cells (*Figure 6*). PV basket cells are the most numerous class of cortical inhibitory cells

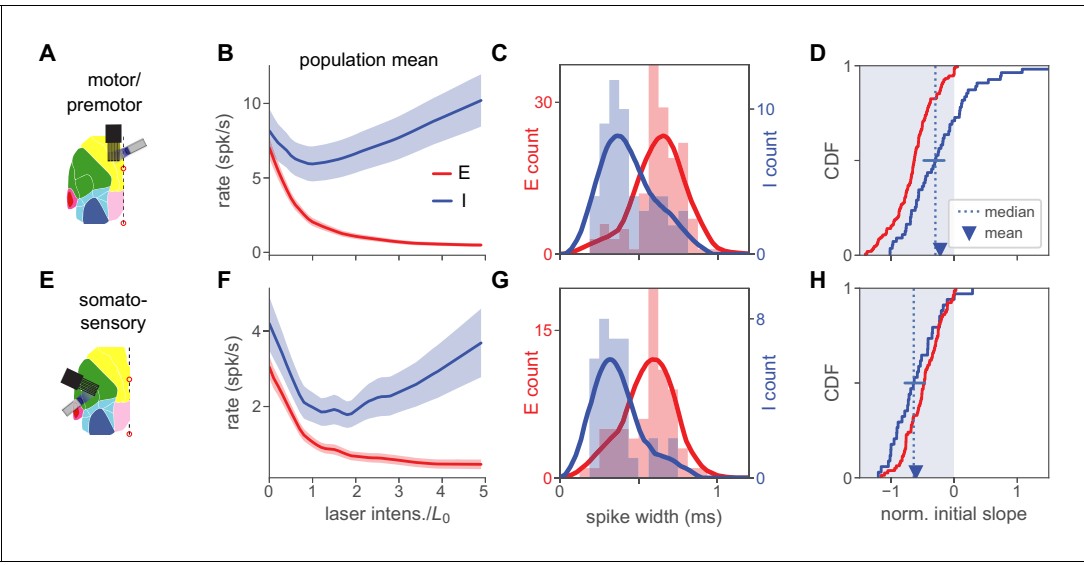

**Figure 5.** Inhibition stabilization across cortical areas. (**A**) Motor/premotor cortex recordings (see *Figure 1—figre supplement 2* for recording locations). (**B**) Motor cortex population firing rates for E and I units. Initial mean response of inhibitory cells is negative, showing paradoxical suppression. Mean rate is significantly reduced ($p<10^{-4}$, paired t-test, rate at 0 vs rate at $L_0$). (**C**) Spike width distributions for E and I units. Units are classified as E or I here by response at high laser power (Materials and methods), independently of spike width, which nonetheless varies with E or I unit identity. (**D**) Normalized initial slope distributions for all units. Red: E. Blue: I. Both mean and median of initial slopes are negative (paradoxical). Mean I slope is negative ($p<0.01$, t-test). Horizontal bar at I median shows 95% confidence interval calculated by bootstrap. (**E–H**) Same as (**A–D**), but for recordings from somatosensory cortex. In (**G**), highest red bar is truncated for visual clarity (value is 22). Mean I rate (**F**) is sigficantly reduced ($p<10^{-7}$, paired t-test). Mean I slope (**H**) is negative ($p<10^{-10}$, t-test). Horizontal bar in (**H**) shows 95% CI around median slope.

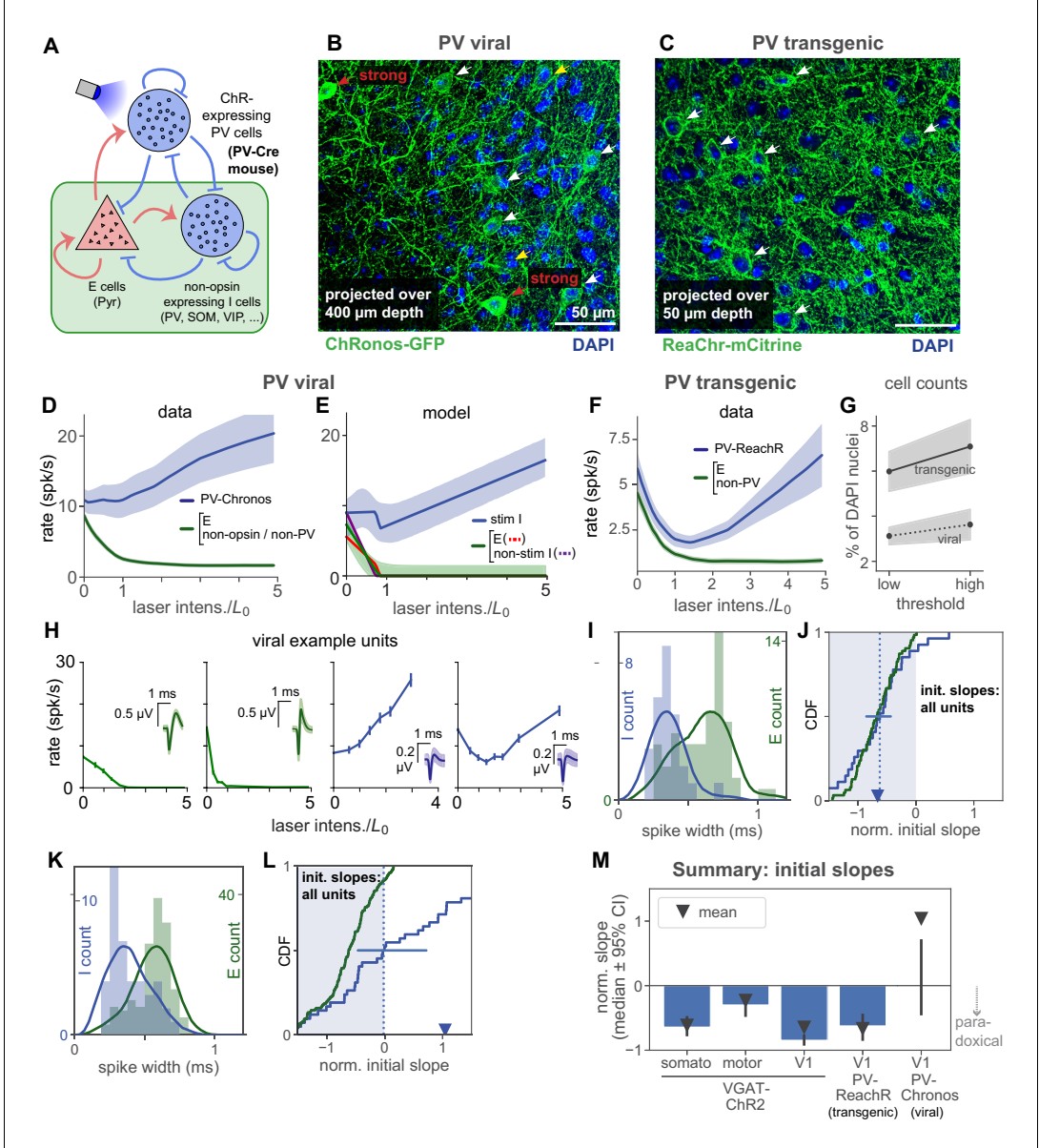

**Figure 6.** Stimulation of parvalbumin-positive inhibitory neurons shows paradoxical effects depend on number of cells stimulated. (**A**) Schematic of cell populations: opsin-expressing PV cells, non-stimulated inhibitory cells (non-PV inhibitory cells: SOM, VIP, . . ., and non-opsin PV cells), and pyramidal (E) cells. Data are from two experiments: Viral expression of opsin (Chronos) in PV neurons (PV-Cre animals with AAV-FLEX-Chronos-GFP injections), and transgenic expression of opsin (ReaChR) in most or all PV neurons (PV-Cre;ReaChR-mCitrine). Responses shown are steady-state responses after light stimulation (Materials and methods), to avoid differences in opsin kinetics affecting results. (**B,C**) Histological characterization of viral (**B**) and transgenic (**C**) expression in superficial layers. Viral expression shows more variability across neurons (cf. red and white arrows), and fewer expressing cells (viral image is a projection across greater depth than transgenic). (**D**) Responses to stimulation with viral expression. Non-PV-Chronos (E, non-Chronos, or non-PV) cells: green (N = 152). PV-Chronos (blue) cells (N = 42, 21%), identified by responses at high laser power (Results; see *Figure 2—figure supplement 3* for validation against pharmacology-based classification in V1 data). Compared to when all inhibitory cells are stimulated (VGAT-ChR2 mouse line, *Figure 2*), weaker paradoxical suppression is seen (blue line; mean rate is not significantly suppressed: $p>0.05$, paired t-test between rate at 0, rate at $1 \cdot L_0$) as initial response slope is near zero. (**E**) A model with a subset of inhibitory cells stimulated (60%) can recapitulate the data in (**B**). Shaded region: ±1 s.d. via data bootstrap as in *Figure 4*. Other than splitting inhibitory population into two subsets, network parameters (Materials and methods) are as inferred in *Figure 4*. (**F**) Population responses for transgenic expression of ReaChR in PV neurons. Unlike the viral-expression data (**D**), the identified inhibitory cells in these experiments show paradoxical suppression; mean firing rate is significantly suppressed ($p<10^{-6}$, paired t-test on initial slopes). (**G**) Cell counts in histological sections. Solid/dotted black lines: means across two independent human counters; upper and lower gray boundaries give results from each counter. X-axis shows variation as counters were asked to use a high or low threshold for accepting an opsin positive cell. Cells were counted across 400 μm depth and are expressed (y-axis) as a percent of DAPI-positive nuclei. Transgenic

*Figure 6 continued on next page*

*Figure 6 continued*

expression gives about twice as many opsin expressing cells as viral expression, in addition to the differences in expression heterogeneity seen in (**B–C**). (**H**) Example viral (PV-Chronos) units show diversity of responses to stimulation. Some narrow spiking units (blue, rightmost two panels) show non-paradoxical initial increases, and some show paradoxical initial suppression. (**I,J**) Distribution of spike widths (**I**) and initial slopes (**J**) for experiments using transgenic PV-ReaChR (bottom). Same conventions as in *Figure 5*, except here colors are as shown in panel (**A**). (**K,L**) Same as (**I,J**) for viral expression. Although inhibitory units are classified by response at high laser power, differences in spike width are visible in both datasets. Viral mean and median slopes are zero or positive (t-test for negative mean $p>0.05$, Mann-Whitney U for negative median $p>0.05$; 22/42 (53%) negative I slopes); transgenic mean and medians are negative (negative mean $p<10^{-7}$, negative median $p<10^{-7}$; 24/27 (89%) negative I slopes). (**M**) Summary of mean and median inhibitory cell initial slopes for these data and data from V1, somatosensory, and motor cortex.

The online version of this article includes the following figure supplement(s) for figure 6:

**Figure supplement 1.** Additional analysis of inhibitory responses to partial stimulation of inhibitory population.
**Figure supplement 2.** No effect of opsin kinetics on paradoxical effects.

and make strong synapses near the somata of excitatory cells (for review, see *Tremblay et al., 2016*), and PV stimulation effectively suppresses network firing rates (e.g. *Glickfeld et al., 2013*; *Sparta et al., 2014*).

We first used viral methods to express an opsin in PV neurons, injecting a Cre-dependent adeno-associated virus (AAV) encoding an excitatory opsin (Chronos; *Klapoetke et al., 2014*), into a transgenic mouse line (PV-Cre; *Hippenmeyer et al., 2005*). In these experiments, the network neurons can be divided into three populations (*Figure 6A*): (1) excitatory, (2) Chronos-expressing PV inhibitory (PV-Chronos), and (3) remaining inhibitory neurons: non-PV (e.g. somatostatin-positive, etc.) or non-Chronos-expressing PV neurons. We identified PV-Chronos cells by measuring whether cells' firing is increased at high laser intensity (statistically significant increase at maximum laser power, as in *Figure 5*; see Materials and methods). In V1 VGAT-ChR2 experiments, this classification by response at high laser power produces qualitatively similar measures of paradoxical suppression as pharmacology-based classification, *Figure 2—figure supplement 3*. Non PV-Chronos cells (without increase in firing at high laser intensity), are likely either excitatory cells, non-PV, or non-expressing cells. Supporting the idea that our classification approach identifies PV-Chronos cells, the majority of classified inhibitory cells have narrow waveforms (*Figure 6I*). While not all PV-positive cells are basket cells (*Taniguchi et al., 2013*), the large proportion of narrow-waveform units we found suggest our inhibitory classification detects a number of fast-spiking cells.

Stimulating the PV cells produced no significant paradoxical effect on average (*Figure 6D*, blue). From a theoretical point of view, this average inhibitory response is an important measure, as with the standard two-population ISN model, e.g. *Tsodyks et al., 1997*, it is the average response (averaged over inhibitory cells) that is paradoxically suppressed when a network is inhibition-stabilized and strongly coupled. A second important measure, e.g. for experimentalists wishing to identify inhibitory neurons, is how many individual inhibitory cells show paradoxical effects. Examining individual units showed that PV-Chronos units often showed no paradoxical effect. Initial slope was often positive and thus non-paradoxical (e.g. *Figure 6H*; median not sig. dif. from zero; summarized in *Figure 6K–L*; see legend for statistical tests).

One reason why stimulation with viral expression might produce no paradoxical effect on average is that viral expression could target only a subset of the PV cells (Mathematical methods; [*Sadeh et al., 2017*]). We first examined the effect of inhibitory cell subset stimulation in a model, the rate model of *Figure 4*, using network parameters inferred there. Stimulating a subset of inhibitory neurons in the model (60%) reduced the magnitude of the average paradoxical effect (*Figure 6E*, analytical derivation in Materials and methods, and *Figure 6—figure supplement 1*), and described the data (*Figure 6D*) well. We also considered a model with multiple subclasses of inhibitory neurons, using the connectivity structure measured in *Pfeffer et al., 2013*, and found there also that paradoxical effects were not clear when approximately half of the PV cells were stimulated (*Figure 6—figure supplement 1*).

Why, intuitively, can stimulation of a subset of inhibitory cells eliminate paradoxical effects? In networks of strongly-coupled excitatory and inhibitory neurons, increased inhibitory activity produced by stimulation suppresses excitatory activity and results in a withdrawal of recurrent excitation; this, in turn, drives suppression of stimulated cells (the paradoxical effect). When only a fraction of inhibitory cells is stimulated, the withdrawal of excitation coexists with other effects produced by

recurrent interactions. In particular, increased activity of stimulated inhibitory cells tends to suppress non-stimulated inhibitory cells and to produce a withdrawal of inhibition to excitatory cells and to stimulated inhibitory cells. This withdrawal of inhibition increases with the fraction of non-stimulated inhibitory cells and, if large enough, can overcome the withdrawal of excitation and prevent paradoxical suppression of stimulated cells.

To test these models, and experimentally determine whether differences in number of cells expressing opsin could change the paradoxical effect, we used a transgenic approach to express a different excitatory opsin in PV cells (PV-Cre;ReaChR transgenic mice). In these mice, the opsin is expressed in most or all PV cells (*Lin et al., 2013*). (Since we study steady-state firing rates (Materials and methods), we do not expect differences in the onset or offset kinetics of the opsin to affect these measurements, and we verified that considering different time windows of the steady-state response does not change the results, *Figure 6—figure supplement 2*). Because we measured responses for a range of light intensities, differences in viral and transgenic mean levels of opsin expression would not affect paradoxical effects. On the other hand, differences in the number of expressing cells, or variability in opsin levels across cells, are predicted to change the paradoxical effect. Indeed, we found that, with transgenic expression, the average paradoxical effect was present (*Figure 6F*). Further, the fraction of inhibitory cells that showed paradoxical responses was significantly larger in PV-ReaChR transgenic animals than in PV-Chronos animals (*Figure 6J,L*), and was similar to the VGAT-ChR2 data in V1, somatosensory, and motor/premotor cortex (*Figure 6M*).

Changes in the number of stimulated cells could be affected by the virus infecting only a subset of neurons (*Watakabe et al., 2015*), or by the virus yielding different levels of opsin expression in different neurons, so that only a subset are recruited strongly at any given light intensity. To examine differences in expression pattern, we counted neurons in histological sections from both the viral (PV-AAV-Chronos) and transgenic (PV-ReaChR) animals. For these comparisons, the Cre line, expressing Cre in PV neurons, was held constant. We observed both effects. First, viral expression created some cells with strong, and some cells with weak, opsin expression (*Figure 6B–C*). Second, viral expression yielded opsin in about half as many cells as transgenic expression (*Figure 6G*); positive-opsin cell percentage significantly different between viral and transgenic cases, both observers and both thresholds, $p<0.05$, $\chi^2$ test). The differences in the number of opsin-expressing cells roughly matched our model predictions (60% of neurons stimulated, *Figure 6E*).

Taken together, these data and our analysis support the idea that V1 and other areas of the mouse cortex are strongly coupled and operate as an ISN. Yet whether an average paradoxical effect is seen depends on the number of inhibitory cells stimulated. The heterogeneous responses we saw in PV-Chronos experiments (about half of inhibitory cells paradoxical, half non-paradoxical, *Figure 6D,H,L*, also may in part explain why paradoxical inhibitory suppression,and thus ISN operation, has not been more widely reported in optogenetic PV-stimulation experiments (also see Discussion).

## Paradoxical effects are also seen in deep-layer recordings

To this point all the neurophysiological data we have reported is from the upper layers (L2/3 and 4) of the cortex, recorded within 400 μm of the cortical surface. We focused on the superficial layers because the blue light we used to activate ChR2 and Chronos does not penetrate more than a few hundred microns into the tissue (*Yona et al., 2016*), so the inhibitory neurons that receive direct optogenetic input are those in the superficial layers. However, blue light delivered to the cortical surface to stimulate inhibitory neurons has been seen to suppress activity across cortical layers, presumably due to polysynaptic effects (*Li et al., 2019*). Because we recorded data using silicon probes that span most of the depth of the cortex, we could also assess whether light delivered to the top of the cortex also produced paradoxical suppression in deeper inhibitory neurons.

We examined units (*Figure 7*) recorded $\geq$ 500 μm from the cortical surface (Materials and methods). In our multi-area recordings from superficial layers (e.g. *Figure 5*), we classified inhibitory cells in different brain areas based on *increases* in inhibitory response at strong light intensities, when excitatory cells are silent or largely suppressed. But in deep recordings, presumably due to attenuation of stimulation light from scattering and absorption, at our highest stimulation intensities we did not always see clear mean increases in inhibitory firing rates (e.g. *Figure 7B*). Therefore, for these deep recordings we classified neurons as inhibitory based on waveform width. As in superficial layers, we found bimodal distributions of waveform width in our deep-layer data

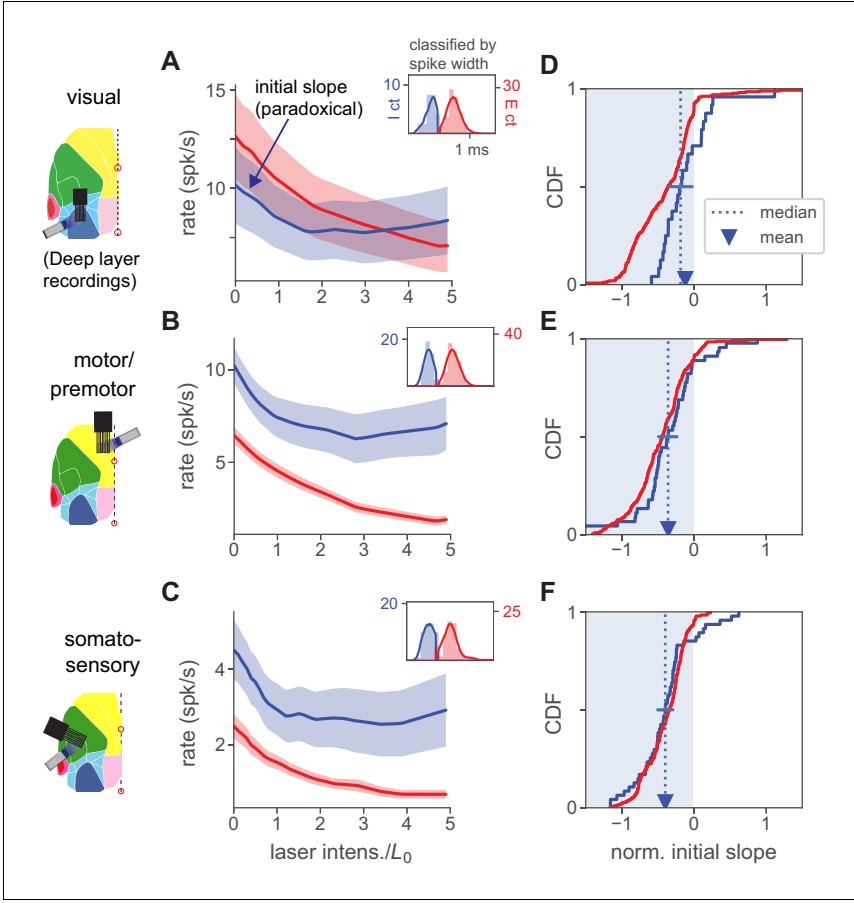

**Figure 7.** Paradoxical inhibitory suppression is also seen in deep-layer recordings. (**A**) Population average responses of deep-layer (recorded ≥500 μm from cortical surface) units, classified as inhibitory (blue) or excitatory (red) by waveform width (inset; solid line is kernel density fit to underlying histogram, expressed in units of counts on they-axes, and spike width in ms on the x-axis, see Materials and methods). Shaded area shows ± 1 SEM about mean. $L_0$ is defined for all units recorded in a single session based on responses to superficial-layer recordings. Initial slope of inhibitory average response is negative (paradoxical). (**B,C**) Same, for motor/premotor and somatosensory. (**D–E**) Initial slopes of all recorded units shown in A-C. Conventions as in *Figure 2K*. As in that panel, slopes here are normalized by baseline rate so that minimum slope is −1. Means and medians of individual inhibitory neurons' slopes are all negative (t-test, V1 $p<0.05$, others $p<0.01$) except for V1 median (blue error bar shows 95% confidence interval around median via bootstrap: upper CI, V1 0.03, motor −0.22, somato −0.33). Also, the population firing rate decrease is significant for all three areas ($p<0.001$, except V1 $p<0.05$, Mann-Whitney U on summed population counts, baseline vs. rate at $L_0$).

The online version of this article includes the following figure supplement(s) for figure 7:

**Figure supplement 1.** Effects of PV stim with viral vs. transgenic expression in deep-layer neurons is similar to effects in superficial layers.

(*Figure 7A–C*, insets). In our upper-layer V1 data, we compared classification of V1 units via pharmacology, waveform width, and responses at high power (*Figure 2—figure supplement 3*). Based on this comparison, in these deep-layer data, we do not expect excitatory cells to be classified as inhibitory (i.e. few or no excitatory units have narrow waveforms), even though some inhibitory cells with wider waveforms will be missed with this approach (i.e. some inhibitory units have wide waveforms).

Examining cells recorded from deeper layers, we found that V1, motor, and somatosensory areas all showed initial paradoxical responses (*Figure 7*). However, the maximal suppression of inhibitory and excitatory cells was not as strong as in superficial layers (*Figure 2*, *Figure 5*). (Supporting the idea that this weaker effect of stimulation is due to attenuation of blue light, paradoxical effects in deep layers were smaller in these data than when using red light to stimulate ReaChR in V1 PV

neurons, *Figure 7—figure supplement 1*.) In all these cases (*Figure 7A–C*), the mean population inhibitory responses showed initial suppression.

Thus, the deep-layer data shows no evidence for a different pattern of responses than seen in the upper layers – deep-layer inhibitory cells show paradoxical suppression to excitatory stimulation. However, because light applied to the cortical surface affects the superficial layer inhibitory neurons most directly, we do not make the claim that these data are final proof that deep layers operate in the ISN regime. On the other hand, the deep-layer data do provide evidence against alternative, non-ISN, models for L2/3 responses, such as the potential alternative where deeper inhibitory cells increase their firing and inhibit L2/3 cells. Instead, we find that most inhibitory cells we record, across layers, are paradoxically suppressed by stimulation.

### No transition out of inhibition stabilization is seen during anesthesia, at lower network firing rates

Theoretical studies have pointed out that a network can switch from non-ISN to ISN when external inputs are increased (*Ahmadian et al., 2013*). If the cortical network did transition out of the ISN state at lower activity levels, this would be computationally important, as the way inputs sum can change depending on whether the network operates as an ISN or not (*Ahmadian et al., 2013*; *Rubin et al., 2015*). Our results above (*Figure 2*, *Figure 3*) show that superficial layers of mouse V1 operate as an ISN at rest (i.e. without sensory stimulation), and also show the transition from ISN to non-ISN is below the level of spontaneous activity, as network activity is decreased by optogenetic stimulation of inhibitory neurons. (The transition point is where the slope of I rates vs stimulation intensity switches from being negative to positive.) However, we wished to determine whether a brain state with lower activity levels, as that produced by light anesthesia, might result in a transition to a non-ISN regime.

Under light isoflurane anesthesia, we found no evidence of a transition, and instead found that paradoxical inhibitory suppression was maintained (*Figure 8A*). At 0.25% isoflurane (a low level, as surgical levels are often 1.0% and above), spontaneous firing rates of excitatory neurons are reduced (*Figure 8C*; mean 6.4 spk/s reduced to 3.5 spk/s, $p<0.02$, Wilcoxon signed-rank test; inhibitory neurons' rates show a negative trend but are not statistically different, 14.5 spk/s awake to 11.6 spk/s anesth., $p<0.10$, Wilcoxon signed-rank tesk). Thus, the network changes induced by anesthesia do not cause the network to transition out of the ISN state. At this low level of anesthesia, we did not observe prominent up and down state slow oscillations. In one experiment, we used a higher level of anesthesia (0.5% isoflurane, *Figure 8B*), yielding even lower firing rates but still preserving the paradoxical effect. Further confirming the robustness of coupling to changes with anesthesia, the distribution of response slopes is roughly unchanged (*Figure 8C*), suggesting the network is far from a transition into a non-ISN state. Anesthesia thus preserves the paradoxical inhibitory response, leaving the network still an ISN.

## Discussion

These data show a signature of inhibitory stabilization, the paradoxical suppression of inhibitory cells to optogenetic excitation, in several different areas of mouse cortex, V1, S1, and motor cortex, during spontaneous activity in the absence of sensory stimuli. An important aspect of the work is that we stimulated all inhibitory cells together, using a mouse line in which all inhibitory cells express opsin. When single inhibitory subclasses are stimulated, paradoxical effects do not prove that the network's excitatory cells are unstable without inhibition (Mathematical Methods; also see *Litwin-Kumar et al., 2016*; *Mahrach et al., 2020*). However, our observation of paradoxical suppression when stimulating all inhibitory cells together, supported by a model that describes the data well (*Figure 4*), is evidence that these cortical networks do indeed have strong excitatory recurrent coupling, and do indeed operate in the ISN regime.

We found, as predicted by ISN models, that strong inhibitory stimulation causes the network to transition into a non-ISN state. In this non-ISN state, inhibitory responses are non-paradoxical and increase their firing when stimulated, and excitatory neurons are suppressed. This transition from ISN to non-ISN behavior is a prediction of ISN models. We saw paradoxical suppression of the mean inhibitory firing response in both superficial and deep layers of the three areas, and we found that a clear majority of inhibitory units (except in one of these six measurements, deep layers of V1) show

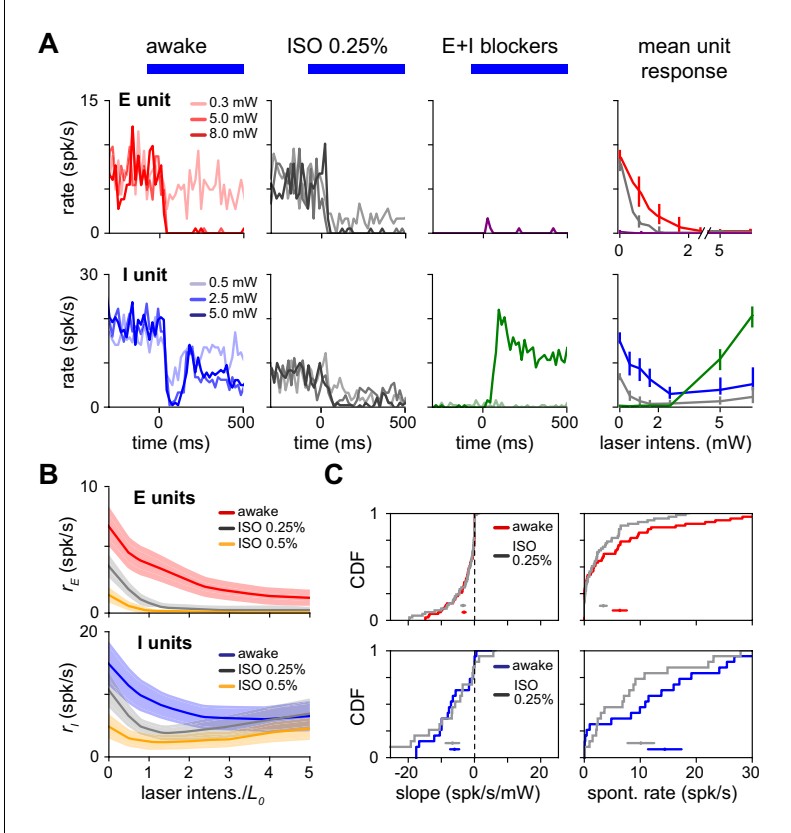

**Figure 8.** Paradoxical response is preserved with lower network activity due to anesthesia. (A) Single excitatory (top) and inhibitory (bottom) unit response in the awake state (no anesthesia, no synaptic blockers), under anesthesia with no blockers (isoflurane, 0.25%) and with synaptic blockers (CNQX, APV, bicuculline; Materials and methods) plus anesthesia. (B) Population average response with (gray) and without (blue;red) anesthesia. Spontaneous firing rates are reduced for both the E (top) and I (bottom) populations below the awake firing rates, yet inhibitory paradoxical suppression is preserved with anesthesia (lower panel: gray line shows negative initial slope). In these experiments, inhibitory responses are weaker at high powers, but this does not affect paradoxical suppression. (C) Distribution of initial slope and spontaneous rate before and after anesthesia. Initial slope is largely preserved while excitatory and inhibitory spontaneous rates are suppressed. Colored horizontal bars within each panel show mean ± SEM for each distribution. Initial slopes are plotted here non-normalized (units spk/s/mW) to show the slopes are quantitatively similar across firing rate changes; normalized slopes are shown in *Figure 8—figure supplement 1*, and their means and medians remain negative across anesthesia state.
The online version of this article includes the following figure supplement(s) for figure 8:

**Figure supplement 1.** Normalized initial slopes for units recorded in anesthesia experiments.

paradoxical suppression when all inhibitory cells are stimulated. We used several ISN rate models to fit data from V1, aided by pharmacological manipulations that yield additional informative data about the network. We find that despite having fewer parameters than data degrees of freedom, the ISN model describes the data well. During light anesthesia, mean V1 network firing rates decrease, but we still observed ISN paradoxical suppression. Finally, we explored the effect of stimulating a subclass of inhibitory neurons with strong network effects: the PV neurons. We find that stimulating PV cells also produces paradoxical effects. We compared viral and transgenic expression strategies and found that viral expression did not produce average paradoxical effects. This viral/transgenic difference is consistent with theoretical observations that changing the fraction of stimulated inhibitory cells changes the strength of the paradoxical effect, and is supported by histological data.

## Multiple lines of evidence rule out other circuits and argue for the strongly-coupled ISN explanation

Our data and theoretical results provide the strongest evidence currently available to rule out non-ISN interpretations. Three prominent features of our data supporting a strongly-coupled recurrent origin for the results are: (1) that inhibitory response dynamics match ISN predictions (*Figure 3*), (2) that an ISN model describes the data well and makes predictions verified in the data (*Figure 4*), and (3) that many inhibitory cells show paradoxical effects at low stimulation intensity but increase their firing rate at higher stimulation intensity (e.g. *Figure 2*, *Figure 5*).

These observations allow us to rule out several potential alternatives to the ISN explanation for the paradoxical effects we observed. One potential alternative might be a model with multiple inhibitory populations and weak recurrent coupling, perhaps driven to fire spontaneously by strong external input such as thalamic or corticocortical input. In principle, paradoxical inhibitory responses could be produced in such a non-ISN model via feedforward, inhibitory-onto-inhibitory,

interactions. Specifically, in this alternative, stimulation of one inhibitory subclass (e.g. PV cells) would inhibit a second set of inhibitory cells whose firing rate would decrease (e.g. somatostatin or SOM cells; the same explanation could hold with the PV and SOM classes swapped). However, in a feedforward inhibitory model, there would be little reason to predict inhibitory neurons to change from decreasing to increasing rate as stimulation intensity increases. ISN models, on the other hand, are predicted to transition into a non-ISN state for strong inhibitory stimulation where inhibitory cells begin to increase their firing, as our data show (e.g. *Figure 2L*).

The large fraction of inhibitory cells that we found to be suppressed (paradoxical) at low stimulation intensity (*Figure 2J–K*) further argues against alternative models with paradoxical effects

created by multiple differentially-responding inhibitory cell populations. While excitatory opsins expressed in inhibitory neurons can sometimes cause synaptic release at terminals (*Zhao et al., 2011*; *Babl et al., 2019*), we found that many inhibitory cells fired in response to optogenetic stimulation in the presence of E and I blockers, showing that our stimulation did produce substantial neural spiking. Further, ISN-generated paradoxical effects are network effects that rely on inhibitory cells affecting other local neurons, so even if stimulation did cause terminal release, this should not affect our conclusions. Finally, ISN models predict, and we observed, that the transition point from inhibitory suppression to excitation occurs when excitatory cells are largely suppressed. It might be possible for that to occur in a non-ISN case, such as in the feedforward scenario, but it would require fine-tuning of parameters, and our ISN model's accurate description of the pharmacology data makes such fine tuning even more unlikely.

Other non-ISN scenarios would also rely on feedforward inhibitory effects made unlikely by our data. For example, alternative models could have feedforward inhibition arising from rare, powerful inhibitory cells, perhaps so rare that we could not record them, or input from deeper-layer inhibitory cells. These feedforward inhibition scenarios are also argued against by the observations outlined above (e.g. the transition from suppression to excitation showed by our recorded inhibitory cells, etc.). And our recordings across depth showed that a majority of inhibitory cells' firing rates were paradoxically suppressed by stimulation, independent of the depth of recording, making it unlikely that our effects were produced by feedforward inhibition arising from a deeper cortical layer. The ISN model remains a consistent explanation of all the data we observed.

## Potential explanations why prior evidence for paradoxical suppression has been mixed

In the last decade, multiple experiments have performed optogenetic stimulation of cortical interneurons, but these studies have yielded mixed results on inhibition stabilization. In layer 2/3 of mouse V1, optogenetic excitation of PV cells with viral expression was shown to generate non-paradoxical modulation of inhibition (*Atallah et al., 2012*). In mouse auditory cortex (A1), in contrast, *Kato et al., 2017* found that suppression of inhibitory cells with viral ArchT produced paradoxical effects in intracellular currents. ISN models make similar predictions for somatic intracellular currents as for the firing rates we measured: if a network is strongly coupled and thus inhibition-stabilized, stimulating all inhibitory neurons will yield paradoxical changes in inhibitory current, as well as a transient increase in inhibitory current before suppression (*Ozeki et al., 2009*; *Litwin-Kumar et al., 2016*). Another A1 study observed mixed paradoxical effects but linked them to a non-ISN mechanism. *Moore et al., 2018* found that stimulation of PV cells using viral transfection produced paradoxical effects in some PV cells and not others. and they reported evidence for feedforward, non-ISN, mechanisms in L2/3 via L4 inhibitory input. In somatosensory and motor cortex, a recent study observed paradoxical suppression for PV subclass stimulation in some layers and not others (*Mahrach et al., 2020*). Finally, many experiments have used excitation of inhibitory cells (e.g. *Glickfeld et al., 2013*) to suppress excitatory activity, without reporting paradoxical inhibitory effects, although a recent survey of such methods does report paradoxical effects (*Li et al., 2019*) in somatosensory and motor cortex.

Our results, combining theoretical analyses with a range of experimental conditions, suggest some explanations why paradoxical inhibitory suppression has not been previously widely reported. First, it appears a significant fraction of inhibitory cells must be stimulated to produce large paradoxical changes in firing rate (*Sadeh et al., 2017*). Stimulating PV cells with viral methods (which can yield subsets of cells with weaker or no expression; see also *Sadeh et al., 2017*; *Gutnisky et al., 2017* for a similar explanation for non-paradoxical effects with viral expression), produced in our hands mixed effects on single cells and weaker mean paradoxical effects (*Figure 6*). Further, stimulating subtypes of inhibitory cells even with transgenic expression can produce different inhibitory responses compared to stimulating all inhibitory neurons (*Mahrach et al., 2020*). Another factor which can, in principle, generate non-paradoxical effects even when the network is an ISN is reducing the area of tissue illuminated by stimulation light. This reduces the number of inhibitory cells stimulated and therefore produces a situation similar to stimulating a subset of network inhibitory cells. However, this effect of reducing optogenetic light area seems to apply mainly for fairly small optogenetic light spots. *Sadeh et al., 2017* found via modeling that paradoxical effects were diminished when illumination spot sizes fell to around 100 μm, well below the 500+ μm diameter spots we

used here to reveal ISN-related paradoxical suppression. This effect of illumination area may, however, impact experiments using optical fibers, which can have small diameters. Beyond the fraction of neurons stimulated, a second explanation for differences, especially in experiments using extracellular recording, is that it can be difficult to definitively identify inhibitory cells. Some inhibitory cells' waveforms are broad (*Figure 2*, *Figure 2—figure supplement 8*), and in an ISN, excitatory and inhibitory cells' firing rate changes are often similar (inhibitory neurons may only show increased activity for a few milliseconds before paradoxical suppression begins). Third, for strong enough drive to inhibitory neurons, ISNs transition to a non-ISN state. In this non-ISN state with low excitatory activity, tracking of excitatory fluctuations is not needed to stabilize the network, and some past experiments may have used stimulation large enough for parts of the network to enter into this second phase of activity, hiding the signature of inhibition stabilization.

## PV cells may be the main source of stabilizing input

While paradoxical effects in a single inhibitory subclass do not imply the overall network is inhibition stabilized (*Figure 6E*; *Mahrach et al., 2020*), PV neurons may well be the principal providers of stabilizing input. In an ISN, strong bidirectional E-I synaptic coupling allows inhibitory cells to stabilize an unstable excitatory network, by tracking and responding to changes in local excitation. Thus, one major parameter that would predict which inhibitory cells would show paradoxical effects is the strength of their synaptic influence on other network neurons. PV neurons receive input from many diverse local excitatory cells (*Bock et al., 2011*) show responses that reflect an average of local excitatory responses (*Sohya et al., 2007*; *Kerlin et al., 2010*), cause network instability when strongly suppressed (an effect which does not result from SOM suppression) (*Veit et al., 2017*), and target virtually all nearby excitatory cells with strong peri-somatic synapses (*Fino and Yuste, 2011*; *Packer and Yuste, 2011*). Other cell types of the diverse inhibitory classes in the cortex (e.g. SOM or VIP+ interneurons; *Tremblay et al., 2016*) also could contribute to stabilization (*Sadeh et al., 2017*), though if their effect on the whole local network is less strong than PV cells, they may be unable to produce paradoxical suppression when stimulated alone.

## Our data support 'loosely balanced' cortical models, and show evidence for ISN operation even at rest

Several influential models of cortical function include inhibition-stabilized parameter regimes, and our data provide new constraints on such models. The 'balanced network' model (*Amit and Brunel, 1997*; *van Vreeswijk and Sompolinsky, 1996*) predicts that both excitatory and inhibitory inputs to single neurons should be large but approximately cancel each other, leading to a balance between excitation and inhibition. This scenario accounts in a parsimonious way for multiple ubiquitous properties of activity in cortex, such as the highly irregular nature of neuronal firing, and the broad distributions of firing rates across neurons. Balanced network models, however, can vary in their recurrent coupling strength, and thus the size of the total excitatory and total inhibitory currents. The analytical work of van Vreeswijk and Sompolinsky (*van Vreeswijk and Sompolinsky, 1996*; *van Vreeswijk and Sompolinsky, 1998*) was performed in the limit of very large numbers of synaptic inputs per cell ($K$), leading to very large E and I currents (whose leading order in $K$ cancel each other — that is, the sum of the E and I currents is small while the sum's variability can be substantial). Other balanced network models (*Amit and Brunel, 1997*; *Brunel, 2000*) used finite, moderate-to-large coupling strength. The network parameters that best fit our data produce moderately large (i.e. within an order of magnitude of threshold) excitatory and inhibitory inputs (*Figure 4*), which are on the same order of magnitude as the ones used in *Brunel, 2000*. This moderate recurrent coupling, or 'loose balance', is also consistent with the SSN model (for review, see *Ahmadian and Miller, 2019*).

The SSN also makes a specific prediction about the ISN regime that our data constrains. The SSN predicts that cortical networks are inhibition-stabilized for strong inputs, but not an ISN for sufficiently weak inputs. Thus, the SSN predicts a transition between two operating regimes as network activity increases. Finding this transition point is important for understanding computation, as the two regimes show different (supralinear or sublinear) modes of input summation. Prior ISN studies (*Ozeki et al., 2009*) have left open whether this transition occurs only with sensory stimulation. A study of responses to two combined sensory stimuli (*Britten and Heuer, 1999*) found a linear-to-sublinear transition for increasing strength (contrast) of visual stimuli, but did not resolve whether

the non-ISN to ISN transition was above or below the level of spontaneous activity. Our data do resolve this, showing several cortical networks are in the ISN state even without sensory stimulation. But a second question our data address is whether during spontaneous activity the network is on the edge of this transition point, or well into the ISN regime. Our work gives two pieces of evidence that without sensory activity the network is far above a transition into a non-ISN state: first, excitatory rates must be substantially suppressed before inhibitory responses switch from paradoxical suppression to firing rate increases (e.g. *Figure 2E*), and second, under light anesthesia, we find network activity is lowered but ISN behavior is preserved. Therefore, our data provides support for the idea that cortical areas generally operate above this transition point, and that the weakly-coupled, non-ISN regime predicted by the SSN is not a regime mouse cortex commonly enters during normal processing in awake behavior.

## Conclusion

Together, the results reported here suggest that inhibition stabilization is a ubiquitous property of cortical networks. This data is consistent with a network that is in the strong coupling regime, but is not extremely strongly coupled as some balanced network models would predict (*Amit and Brunel, 1997*; *van Vreeswijk and Sompolinsky, 1996*). It is tempting to speculate that, while strong coupling allows the network to perform non-trivial computations on its inputs, increasing coupling further might not be optimal for several reasons. First, in the strong coupling limit, only linear network responses are available to a balanced network (unless synaptic non-linearities are included, *Mongillo et al., 2012*), while networks with moderate coupling can combine inputs in a non-linear fashion (*Ahmadian et al., 2013*). Second, maintaining strong connections might be metabolically expensive, so that a moderately-coupled network might represent an ideal compromise for cortical computations.

# Materials and methods

## Experimental methods

### Animals

All procedures were conducted in accordance with the guidelines of the National Institutes of Health. Seven VGAT-ChR2 (*Zhao et al., 2011*; ChR2 targeted at the *Slc32a1* locus), three PV-Cre (*Hippenmeyer et al., 2005*; Cre targeted at the *Pvalb* locus), and two PV-Cre;ReaChR (*Lin et al., 2013*; ReaChR targeted at the *Gt(ROSA)26Sor* locus) mice were used (JAX stock n 014548, 008069, and 024846; 2 females and 10 males; singly housed on a reverse light/dark cycle).

### Cranial window implants

Mice were implanted with a titanium headpost and a transparent window (optical glass, 0.8 mm thickness, 3 or 5 mm diameter) in the left cerebral hemisphere. The windows provided access to the primary visual (V1) and somatosensory cortex, or motor cortices, for imaging and silicon neural probe recordings with optogenetics. Mice were given dexamethasone (3.2 mg/kg, i.p.), 2 hr before surgery. Animals were anesthetized during surgery with isoflurane (1.0%–4% in 100% O$_2$). Using aseptic technique, a headpost was affixed using Metabond (Parkell), and a 3 or 5 mm diameter craniotomy made.

### Hemodynamic intrinsic imaging

To determine the location of V1, we delivered small visual stimuli to animals at different retinotopic positions and measured changes in the absorption of 530 nm light resulting from cortical hemodynamic responses (*Ma et al., 2016*). We evenly illuminated the brain with a 530 nm fiber coupled LED (M530F2, Thorlabs) passed through a 532 nm-center bandpass filter (Thorlabs). Images were collected on a stereo microscope (Discovery V8, Zeiss) through a green long-pass emission filter using a 1x objective (PlanApoS 1.0x, Zeiss) onto a Retiga R3 camera (Q Imaging, captured at 2 Hz with 4 × 4 binning). For retinotopic mapping we presented upward-drifting square wave gratings (2 Hz, 0.1 cycles/degree) masked with a circular window (10° diameter) for 5 s with 10 s of mean luminance preceding each trial. Stimuli were presented in random order at four positions in the right monocular field of view. The response to a stimulus was calculated as the fractional change in intensity between

the average of the 10 frames immediately preceding the stimulus (as baseline) and a 6–10 frame window 1 s after stimulus onset (as response) to match the timecourse of the hemodynamic response (*Chen-Bee et al., 2007*; *Heimel et al., 2007*).

## Viral injections

We used viral injections to express Chronos (*Klapoetke et al., 2014*) in PV-Cre animals. Mice were anesthetized (isoflurane 1–1.5%) and the cranial window implant removed. We used a stereotaxic injection system (QSI, Stoelting Inc) to deliver 500 nL of a 10:1 mixture of AAV1-hSyn-FLEX-Chronos-GFP (UNC Vector Core Stock) and 100 µM sulforhodamine (SR101, Invitrogen, for visualization) at a depth of 200–400 µm. We made 3 to 4 injections spaced 0.5–1.0 mm apart to cover the visual cortex. After the injections a new cranial window was affixed. Viral expression was monitored over the course of days by imaging GFP fluorescence, and allowed to develop for greater than 4 weeks before electrophysiological recordings.

## Electrophysiological recording with optogenetic activation of inhibitory neurons

For recording experiments, we first affixed a 3D printed ring to the cranial window to retain fluid. With the ring in place, we removed the cranial window and flushed the craniotomy site with sterile normal saline to remove debris. Kwik-Sil silicone adhesive (World Precision Instruments) was used to seal the craniotomy between recording days. Using a stereomicroscope on an articulating arm we positioned the end of an optical fiber (600 µm diameter, Doric Lenses) fitted with a light-tight coupler to an optical cannula (400 µm diameter, Thorlabs CFMLC14L02) over our target cortical area, with a slight 10–30° angle from vertical to provide space for the electrodes. We used fiber coupled LED light sources (M470F3for Chronos and ChR2, or M625F2 for ReaChR, Thorlabs) to deliver illumination with peak at 470 nm or 625 nm to the brain. We calibrated total intensity at the entrance of the cannula using a power meter and photodiode (meter model 1918R, photodiode model 918D-SL-OD3R, Newport Corp). The cannula distance from the dura was adjusted to provide a light spot with a full width at half maximum intensity of 0.8–1.2 mm (as measured with a small digital macro documentation microscope, Opti-TekScope); thus, our reported spot irradiance in $mW/mm^2$ is within a factor of two of the reported power (mW). We targeted a multisite silicon probe electrode (Neuro-Nexus; 32-site model $4 \times 8$–100–200–177; four shank, eight sites/shank; sites were electrochemically coated with PEDOT:PSS [poly(3,4-ethylenedioxythiophene): poly(styrenesulfonate)], *Xiao et al., 2006*) to the center of the LED spot using a micromanipulator (MPC-200, Sutter Instruments). With the fiber optic cannula and electrode in place, we removed the saline buffer using a sterile absorbent triangle (Electron Microscopy Sciences, Inc) and allowed the dura to dry for 5 min. After insertion, we waited 30–60 min without moving the probes to reduce slow drift and provide more stable recordings. We isolated single and multiunit threshold crossings (three times RMS noise) by amplifying the site signals filtered between 750 Hz and 7.5 Khz (Cerebus, Blackrock microsystems). During recordings, animals were awake and passively viewing a gray screen. To keep animals awake and alert, animals were water-scheduled (*Histed and Maunsell, 2014*), and a 1 µl water reward was randomly provided on 5% of the stimulus trials; we verified animals were licking in response to rewards during the experiments. Superficial units were those recorded from a site within 400 µm of the cortical surface, identified by monitoring spike and local field potential activity (LFP) on sites at different depths; we typically observed desynchronized cortical LFP activity on any site below the surface and found the first substantial unit activity 100 µm below the surface. Deep units were those recorded 500–800 µm below the surface. Optogenetic stimuli were square light pulses with 2 ms linear ramps at start and end to reduce recording artifacts. Pulses were on for 600 ms and off for 1000 ms at a range of power levels (0.3–10 mW), presented in random order, with 100 repetitions per power level. For population plots, we normalized the light intensity in each experiment to control for fluctuations across experiments due to e.g. changes in dural thickness or tissue light absorption. We found a minimum value of the inhibitory responses ($L_0$, with one value for each experiment, used for each unit recorded in that experiment) by fitting a 2-segment piecewise-linear function to the average inhibitory response (*Figure 2—figure supplement 6*). To avoid biases due to light attenuation at deep sites, $L_0$ was computed for each experiment on the superficial sites only, and this value was

used for the deep site data (*Figure 7*). Population firing rate std. dev. and SEM (*Figure 3*) were calculated on the sum of all unit counts in a given time bin.

## Pharmacological blocking of excitatory and inhibitory synapses

To classify cells using pharmacological weakening of cortical synapses, we divided the experiment into three phases, each 30–45 min in duration: first no blockers, then excitatory blockers, then excitatory plus inhibitory blockers. For each section, we delivered the same optogenetic stimulation protocol (above). To apply the pharmacological agents, we made a hole near the recording electrodes in the agarose on top of the brain, removed the normal saline covering the agarose and hole, and replaced it with a solution containing the agent in saline. After 15 min, the initial application of blocking solution was removed via aspiration and refreshed. We waited a total of 20 min for the pharmacological agent(s) to take effect before recording. Excitatory synaptic blockers (2 mM CNQX, 6 mM APV) affected AMPA, kainate, and NMDA synapses. Inhibitory blockers (1 mM bicuculline) affected GABA-A synapses. Source: Sigma-Aldrich (#C239, #A8054, and #14343).

## Histology and electrophysiology probe tracking

To determine the location and depth of recording electrodes, we coated the shanks with 1,1′-Dioctadecyl-3,3,3′,3′-tetramethylindocarbocyanine perchlorate (DiI) (Thermo Fisher #D282, 50 mg/mL solution in ethanol), prior to insertion into the brain. We dipped the electrode tips 5 times into the DiI solution and allowed the coating to dry for 30 s between immersions. Fluorescent electrode tracks were then visualized in coronal sections of fixed brain tissues. Mice were anesthetized with isoflurane and injected intraperitoneally with pentobarbital sodium (150 mg/kg). They were then perfused transcardially with cold (4°C) PBS followed by cold 4% paraformaldehyde. Brains were extracted and fixed in 4% paraformaldehyde for 6–12 hr and then cryoprotected in a 30% (w/v) sucrose solution in PBS until they sank. Brains were sectioned at 50 μm on a freezing microtome (Leica), mounted on glass slides, and coverslipped with mounting media containing DAPI (Fluoromount-G with DAPI, Electron Microscopy Sciences). Slides were imaged using an Olympus slide-scanner (Olympus BX61VS, Japan).

## Counting opsin-expressing cells

To quantify opsin expression differences between viral (PV-Cre,AAV-Chronos-YFP) and transgenic (PV-Cre::ReaChR-mCitrine) animals, we used a confocal microscope (Zeiss LSM780) to image fluorescent neurons in L2/3 using 50 μm thick coronal brain tissue sections. (We used the same Cre line for both cases, so that any Cre expression patterns would affect neurons in both groups, and >90% of cortical PV neurons are reported to express Cre in this line [*Hippenmeyer et al., 2005*]). For each animal, we chose four 100 μm x 100 μm areas in layer 2/3, and constructed 3D stacks by aligning images across adjacent sections. 3D volumes from 3 viral and three transgenic animals were anonymized and manually counted by two independent observers. Observers were instructed to count cells once with a high threshold for accepting a cell as positive, and once with a low threshold for accepting a cell. Percentages of opsin-expressing cells were calculated as the average number of fluorescently labeled cells divided by the number of DAPI-labeled nuclei.

## Anesthesia during electrophysiology

To test the effects of lowering overall activity on the stability of the cortical network we fitted animals with an isoflurane inhalation mask system (model V-1, VetEquip Inc) and provided anesthesia (0.25–0.5% isoflurane in 100% $O_2$) to put the animal into a lightly anesthetized but awake state during recordings. At the lower 0.25% isoflurane concentration, recordings in V1 displayed no synchronized oscillatory Up/Down state activities and the animals eyes were open throughout. The anesthesia experiments were divided into four sections: awake animal, anesthesia with no blockers, anesthesia with excitatory blockers, anesthesia with inhibitory blockers. Anesthesia was delivered for 20 min before starting recording.

## Spike sorting

Spike waveforms were sorted after the experiment using OfflineSorter (Plexon, Inc). Single units were identified as waveform clusters that showed clear and stable separation from noise and other

clusters, unimodal width distributions, and inter-spike interval histograms consistent with cortical neuron absolute and relative refractory periods. Multiunits were clusters that were distinct from noise but did not meet one or more of those criteria, and thus these multiunits likely group together a small number of single neurons. Signal-to-noise ratio (SNR) (*Kelly et al., 2007*; *Histed, 2018*) of single unit waveforms (median ± std): in visual cortex, 3.6 ± 1.7 (N = 167), see *Figure 2—figure supplement 2*; in other datasets: motor, 3.7 ± 1.2 (N = 103); somatosensory, 3.7 ± 1.7; in V1 PV data (*Figure 6*) 2.91 ± 1.3. Only single-units were analyzed; multiunits were discarded. We also repeated our single-unit analyses using both more- and less- stringent criteria to qualify a unit as a single unit, and found no qualitative differences in the results (*Figure 2—figure supplement 2*). Supporting the idea that single units did not group together multiple units, we found no significant correlation between SNR and baseline firing rate in any of the four datasets (all $p > 0.2$ by linear regression).

## Mathematical methods

### Model equations

To quantitatively analyze the network response, we used a two population rate model (*Wilson and Cowan, 1972*), in which the average firing rates of excitatory (E) and inhibitory (I) populations ($r_E$ and $r_I$, respectively) evolve according to

$$\begin{cases} \tau_E \frac{dr_E}{dt} &= -r_E + \phi_E(W_{EE}\ r_E - W_{EI}\ r_I + I_{EX}) \\ \tau_I \frac{dr_I}{dt} &= -r_I + \phi_I(W_{IE}\ r_E - W_{II}\ r_I + I_{IX} + \lambda L) \end{cases} \tag{1}$$

where $\phi_A$, $I_{AX}$ and $\tau_X$ are the static transfer function (f-I curve), external input and time constant of population $A$ (=E,I), respectively, while $W_{AB}$ is the strength of connections from population $B$ to $A$. The optogenetic stimulation is described by the parameters $\lambda$ and $L$ which represent the efficacy and the intensity of the stimulation light. Results shown in the main text (*Figure 4*) have been obtained with a rectified-linear transfer function

$$\phi_A(x) = a_A [x - x_{0A}]_+ , \tag{2}$$

which is zero for input $x$ smaller than the threshold $x_{0A}$, and increases linearly, with a gain $a_A$, otherwise. This transfer function is the simplest one that can describe the data, which shows an approximately piece-wise linear dependence of firing rates on stimulation intensity (see *Figure 4*). We also fit the data using a different transfer function

$$\phi_A(x) = b_A \log \left\{ 1 + \exp \left[ \frac{a_A}{b_A} (x - x_{0A}) \right] \right\} \tag{3}$$

that smoothes the threshold non-linearity of the rectified linear function, using an additional parameter $b_A$ that controls the width of the exponential region around threshold. This function reduces to the rectified-linear transfer function when $b_A \to 0$. We find that the nonlinear transfer function provides a minor improvement in describing the data and does not significantly affect the values of the inferred parameters (*Figure 4—figure supplement 1*).

In the experiments, recordings of the response are done in three separate phases: (1) A 'normal' phase in which recurrent interactions are intact; (2) A phase with E synaptic blockers; and (3) A phase with both E and I synaptic blockers. The addition of blockers in the model is described with two parameters, $\epsilon_{E,I} \in [0, 1]$, representing the decrease in strength of excitatory and inhibitory synapses. After the addition of excitatory blockers, connectivity is modified as

$$W_{EE}, I_{EX}, W_{IE}, I_{IX} \to \epsilon_E W_{EE}, \epsilon_E I_{EX}, \epsilon_E W_{IE}, \epsilon_E I_{IX} , \tag{4}$$

and with inhibitory blockers, connectivity is modified as

$$W_{EI}, W_{II} \to \epsilon_I W_{EI}, \epsilon_I W_{II} \tag{5}$$

Combining *Equations 1, 4 and 5* with the transfer function of *Equation 2*, the model uses 15 parameters (2 time constants, 2 thresholds, 2 gains, 4 connectivity strengths, 2 external inputs, 1 stimulation efficacy, 2 blocker efficacies) to describe simultaneously the three phases of the experiment.

To compare the model response with the data, we find the equilibrium solution of *Equation 1*, shown in *Equation 6*. Since $\tau_E$ and $\tau_I$ do not affect the equilibrium solution, the number of relevant independent variables in the model reduces to 13. Moreover, the values of $a_E$ and $a_I$ can be reabsorbed in the definitions of $W$, $I$ and $\lambda$. It follows that all the parameters can be inferred up to a proportionality constant and, without loss of generality, we can fix $a_E = a_I = 1$. This reduces the number of independent parameters to 11.

Note that this number of parameters is smaller than the number of parameters needed for piecewise-linear fits of the data (average firing rates vs light intensity for both E and I neurons). Such piecewise-linear fits of the data need 15 parameters: 3 phases times five parameters per phase (three for inhibitory neurons - two per linear region, minus one for the continuity constraint; and two for excitatory neurons, for the single linear region at low intensities). Thus, the model does not have enough parameters to fit successfully the data, unless the model structure accurately describes the cortical responses. Indeed, *Figure 4* shows that the model gives a good description of the data. The additional 4 degrees of freedom present in the data lead to four parameter free model predictions described in the main text.

The fixed point solutions of *Equation 1* (without blockers) are

$$
\begin{cases}
r_E = \frac{(W_{II}+1)(I_{EX}-x_{0,E})-W_{EI}(I_{IX}+\lambda L - x_{0I})}{W_{EI}W_{IE}-(W_{II}+1)(W_{EE}-1)} \\
r_I = \frac{W_{IE}(I_{EX}-x_{0,E})-(W_{EE}-1)(I_{IX}+\lambda L - x_{0I})}{W_{EI}W_{IE}-(W_{II}+1)(W_{EE}-1)}
\end{cases}
\text{and}
\quad
\begin{cases}
r_E = 0 \\
r_I = \frac{I_{IX}+\lambda L - x_{0,I}}{W_{II}+1}
\end{cases}.
\tag{6}
$$

For every value of the laser intensity, the excitatory and inhibitory nullclines are defined as the functions $r_I = r_I(r_E)$ which solve the first and the second line of *Equation 6*, respectively. Every intersection between the two nullclines is a solution of *Equation 6* and gives a stationary state of the network dynamics.

*Equation 6* has two dimensions that can be inferred only up to a multiplicative constant:

$$
\begin{cases}
W_{EE}-1, W_{EI}, I_{EX}, x_{0,E} \to \gamma_E(W_{EE}-1), \gamma_E W_{EI}, \gamma_E I_{EX}, \gamma_E x_{0,E}, \\
W_{IE}, W_{II}+1, I_{IX}, x_{0,I}, \lambda \to \gamma_I W_{IE}, \gamma_I(W_{II}+1), \gamma_I I_{IX}, \gamma_I x_{0,I}, \gamma_I \lambda
\end{cases}.
\tag{7}
$$

Because of these invariances, using data from a single experimental phase, only ratios between parameters can be inferred. This is no longer true when the three phases are considered; in this case all the 11 relevant model parameters can be found.

## Parameter inference

Given the dataset of excitatory and inhibitory responses, the best set of parameters describing the data is found via global optimization as follows:

1. Select random initial parameters in the interval [0,10].
2. Find the optimal set of parameters through a least-squares optimization (python function 'curve_fit', variables are constrained to be positive and, for $\epsilon_{E,I}$, in the interval [0,1]) of the difference between observed rates and model predictions given by *Equation 6*.
3. Repeat the procedure $10^4$ times, and select the solution with minimal error as the optimal parameter set describing the data.

The optimization procedure is applied simultaneously on data from all phases and recording sessions in V1. For the first phase (awake), data from all recording sessions from VGAT-ChR2 animals in V1 (9 days, four animals) were pooled together for fitting. For the second and third phases (E blockers, E+I blockers), in order to describe the variability from day to day of the efficacy of the blockers, we use a different set of $\epsilon_{E,I}$ for each recording session (6 days, two animals). For these two phases, three out of the nine recording sessions are excluded from the analysis because blockers were added after anesthetizing the animals. The optimal parameters found by this approach are:

$$
\begin{aligned}
W_{EE} = 2.56, \ W_{EI} = 1.77, \ I_{EX} = 8.51\,spk/s, \ x_{0E} = 1.19\,spk/s, \\
W_{IE} = 8.54, \ W_{II} = 7.11, \ I_{IX} = 34.16\,spk/s, \ x_{0I} = 8.65\,spk/s, \ \lambda = 6.3\,spk/s
\end{aligned}
\tag{8}
$$

These parameters are used in *Equation 6* to generate the best model description of the data in the first phase of the experiment (*Figure 4A*, first column, black line). For the second and third phases, predictions also depend on the efficacy of synaptic blockers, which are different in the

different recording sessions (see *Figure 4—figure supplement 2*). Data and model predictions for all sessions are shown in *Figure 4—figure supplement 2*; in the main text, we showed averages of the two computed across sessions for each phase (*Figure 4A*, second and third columns, black line).

We use a bootstrap approach, combined with the global optimization described above, to estimate the precision with which model parameters can be inferred from the data. The optimal (minimum-error) estimates obtained from $10^4$ random resamplings (random with replacement) are shown in *Figure 4—figure supplement 2*. Despite the non-uniqueness of the solutions, the bootstrap shows (dashed lines in *Figure 4*: 1 s.d. via boostrap) that model rates are clustered around the data in all three phases of the experiment.

## Stability of the solution

To analyze the stability of the system, we compute the eigenvalues of the dynamical matrix defined by *Equation 1*. For the linear transfer function we find

$$\lambda_\pm = \frac{a_E W_{EI}}{2\tau_E}\left[\left(\frac{a_E W_{EE}-1}{a_E W_{EI}}-r\frac{a_I W_{II}+1}{a_I W_{IE}}\right)\pm\sqrt{\left(\frac{a_E W_{EE}-1}{a_E W_{EI}}+r\frac{a_I W_{II}+1}{a_I W_{IE}}\right)^2-4r}\right] \tag{9}$$

with $r=\frac{\tau_E a_I W_{IE}}{\tau_I a_E W_{EI}}$ This ratio cannot be inferred from the data since $\tau_{E,I}$ do not appear in the static solution. We thus determined $\tau_E$ and $\tau_I$ (shown in *Figure 4E*) from the response dynamics (*Figure 4D*).

## With multiple inhibitory subclasses, ISN operation need not imply paradoxical suppression

To understand how the network response to PV stimulations depends on its structure, here we analyze a three-population model (*Figure 6A*) with one excitatory and two inhibitory populations, referred hereafter as $E$ (pyramidal cells), $P$ (ChRonos-expressing PV cells), and $I$ (non-expressing PV , SOM, VIP...). The network dynamics is described by

$$\begin{cases} \tau_E\frac{dr_E}{dt} &= -r_E + \phi_E(W_{EE}r_E - W_{EI}r_I - W_{EP}r_P + I_{EX}), \\ \tau_I\frac{dr_I}{dt} &= -r_I + \phi_I(W_{IE}r_E - W_{II}r_I - W_{IP}r_P + I_{IX}), \\ \tau_P\frac{dr_P}{dt} &= -r_P + \phi_P(W_{PE}r_E - W_{PI}r_I - W_{PP}r_P + I_{PX} + \lambda L). \end{cases} \tag{10}$$

The model predictions in *Figure 6* have been obtained from *Equation 10* using

$$\begin{aligned}
W_{EE} &= W_{EE}^{VGAT}, & W_{EI} &= (1-frac)W_{II}^{VGAT}, & W_{EP} &= frac W_{II}^{VGAT}, & I_{EX} &= I_{EX}^{VGAT}, \\
W_{IE} &= W_{IE}^{VGAT}, & W_{II} &= (1-frac)W_{II}^{VGAT}, & W_{IP} &= frac W_{II}^{VGAT}, & I_{IX} &= I_{EX}^{VGAT}, \\
W_{PE} &= W_{IE}^{VGAT}, & W_{PI} &= (1-frac)W_{II}^{VGAT}, & W_{PP} &= frac W_{II}^{VGAT}, & I_{PX} &= I_{EX}^{VGAT},
\end{aligned} \tag{11}$$

where $frac \in (0,1)$ represents the fraction of ChRonos-expressing PV cells, $W_{AB}^{VGAT}$ and $I_A^{VGAT}$ represent the parameters of *Equation 8*, obtained using the V1 data from the VGAT-ChR2 mouse line.

Using the rectified linear transfer function of *Equation 2*, and assuming $a_A = 1$ to simplify expressions, the equilibrium response to stimulation of the $P$ population is given by

$$r_P(L) - r_P(0) = \frac{(1-W_{EE})(W_{II}+1)+W_{EI}W_{IE}}{N}\lambda L \tag{12}$$

with

$$\begin{aligned}
N = (1-W_{EE})[(W_{II}+1)(W_{PP}+1)-W_{IP}W_{PI}] \\
-W_{EI}[W_{IP}W_{PE}-W_{IE}(W_{PP}+1)]+W_{EP}[(W_{II}+1)W_{PE}-W_{IE}W_{PI}]
\end{aligned}$$

*Equation 12* shows that, both for $W_{EE}<1$ and $W_{EE}>1$, depending on the network connectivity, the ratio on the r.h.s. can be either positive or negative. It follows that, in a network with two inhibitory populations, an unstable excitatory subnetwork does not imply paradoxical suppression when one of the populations is stimulated. This can be seen more easily in a simplified model in which connectivity depends only on the identity of the presynaptic unit. In particular, using $W_{EE}=W_{IE}=W_{PE}=W$, $W_{EI}=W_{II}=W_{PI}=k_I W$, and $W_{EP}=W_{IP}=W_{PP}=k_P W$, *Equation 12* becomes

$$r_P(L) - r_P(0) = \frac{1 + (k_I - 1)W}{1 + (k_P + k_I - 1)W} \tag{13}$$

In the large $W$ limit, in which the excitatory population is unstable without inhibition, the network shows the paradoxical effect only if $k_I < 1$ and $k_P > 1 - k_I$. (Note that, for the network to be stable in this condition, the matrix of the coefficients of *Equation 10*, linearized around the fixed point, must have eigenvalues with a negative real part, a requirement which also involves the magnitude of $\tau_E$, $\tau_I$, and $\tau_P$.)

## Acknowledgements

Supported by the NIMH Intramural Research Program and by NIH BRAIN U01 NS108683 (to MH and NB). We thank K Miller and J Reynolds for discussions and the NIMH Instrumentation Core for technical support. We also thank L Glickfeld, S Lee and K Miller for comments on the manuscript. This work used the computational resources of the NIH HPC Biowulf cluster (http://hpc.nih.gov).

## Additional information

### Funding

| Funder | Grant reference number | Author |
| --- | --- | --- |
| National Institutes of Health | U01NS108683 | Nicolas Brunel Mark H Histed |
| National Institutes of Health | NIMH Intramural Research Program | Mark H Histed |

The funders had no role in study design, data collection and interpretation, or the decision to submit the work for publication.

### Author contributions

Alessandro Sanzeni, Conceptualization, Data curation, Formal analysis, Validation, Investigation, Visualization, Methodology, Writing - original draft; Bradley Akitake, Conceptualization, Data curation, Supervision, Validation, Investigation, Visualization, Methodology, Writing - review and editing; Hannah C Goldbach, Caitlin E Leedy, Investigation, Methodology, Writing - review and editing; Nicolas Brunel, Conceptualization, Resources, Formal analysis, Supervision, Funding acquisition, Methodology, Project administration, Writing - review and editing; Mark H Histed, Conceptualization, Resources, Data curation, Formal analysis, Supervision, Funding acquisition, Validation, Investigation, Visualization, Methodology, Project administration, Writing - review and editing

### Author ORCIDs

Alessandro Sanzeni https://orcid.org/0000-0001-8758-1810
Bradley Akitake https://orcid.org/0000-0002-1817-4573
Hannah C Goldbach https://orcid.org/0000-0002-5697-4694
Caitlin E Leedy https://orcid.org/0000-0001-9277-5409
Nicolas Brunel http://orcid.org/0000-0002-2272-3248
Mark H Histed https://orcid.org/0000-0001-8235-7908

### Ethics

Animal experimentation: All procedures were conducted in accordance with the guidelines of the National Institutes of Health, and approved by the institutional animal care and use committee (IACUC, protocol UNCB-01) of the NIMH Intramural Program.

### Decision letter and Author response

Decision letter https://doi.org/10.7554/eLife.54875.sa1
Author response https://doi.org/10.7554/eLife.54875.sa2

# Additional files

## Supplementary files

• Transparent reporting form

## Data availability

Data and code to generate plots are available at: https://github.com/histedlab/code-Sanzeni-inhibition-stabilization-cortex (copy archived at https://github.com/elifesciences-publications/code-Sanzeni-inhibition-stabilization-cortex).

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

## Appendix 1

# Paradoxical response implies inhibitory stabilization in networks with short term plasticity

In the main text, we have shown that cortical response to optogenetic stimulations is consistent with the network being stabilized by inhibition. Our conclusion was based on theoretical work linking paradoxical inhibitory responses to inhibitory stabilization of unstable excitatory activity (*Tsodyks et al., 1997*). This work shows that network stability is determined by two variables, strength of recurrent connections and time scale of neuronal responses, which are assumed to be constant in time. Short term plasticity (STP) (*Tsodyks and Markram, 1997*; *Markram et al., 1998*) could potentially modify the link between paradoxical response and inhibition stabilization, as it dynamically modifies synaptic efficacy, but its effects on network stability have not been analyzed yet. In this section, we show that paradoxical inhibitory response implies inhibitory stabilization also when STP is taken into account. However, the reverse is no longer guaranteed to be true, as inhibition stabilization no longer necessarily implies paradoxical response. Therefore, the conclusions of the main text remain valid also in a modeling framework which includes STP.

## 1. Mathematical description of STP

To investigate the effects of STP on the relationship between paradoxical effect and inhibitory stabilization, we use the phenomenological description of STP developed in *Tsodyks et al., 1997*; *Markram et al., 1998*; *Tsodyks et al., 1998*. This framework is accurate enough to capture quantitatively both short term depression (STD) and short term facilitation (STF), and is simple enough to allow analytical investigation. The state of each synapse is described by two variables: the fraction $x$ ($0 \leq x \leq 1$) of vesicles available for release, and the fraction $u$ ($0 \leq u \leq 1$) of available vesicles that release neurotransmitter after a presynaptic spike (release probability). In the model, a presynaptic spike arriving at time $t$ opens calcium channels in the presynaptic terminal, which generates an increases the value of $u(t^-)$ by an amount $U[1 - u(t^-)]$ ($0<U<1$ is a fixed parameter) and produces the release of a fraction $u(t^+)x(t^-)$ of vesicles. After the spike, channels close with a time constant $\tau_F$ and vesicles recover with a time constant $\tau_D$. As shown in *Tsodyks et al., 1998*, when presynaptic spikes are Poisson-distributed with rate $r$, the dynamics of $x$ and $u$ is given by

$$\begin{cases} \frac{du}{dt} = -\frac{u}{\tau_F} + U(1-u)r, \\ \frac{dx}{dt} = \frac{1-x}{\tau_D} - \Omega x r, \\ \Omega = [u + U(1-u)]. \end{cases} \tag{A1}$$

In this model, synaptic efficacy and postsynaptic current are proportional to $\Omega x$ and $\Omega x r$, respectively. The modulation of synaptic efficacy by the presynaptic rate can be understood as follows. When $u$ is fixed, $x$ and the synaptic efficacy decrease with $r$, leading to synaptic depression. When $x$ is fixed, the increase in $u$ with $r$ augments synaptic efficacy and leads to synaptic facilitation. When both $u$ and $x$ are allowed to vary, facilitation dominates at low $r$ and depression dominates at high $r$ (*Markram et al., 1998*; *Tsodyks et al., 1998*). A stationary input rate $r(t) = r_0$ gives

$$\Omega_0 = U\frac{1 + \tau_F r_0}{1 + U\tau_F r_0}, \quad x_0 = \frac{1}{1 + \Omega_0 \tau_D r_0}, \tag{A2}$$

and, since $U<1$, produces a synaptic current which increases monotonically with $r_0$; this monotonic increase will be important in the upcoming sections.

In what follows, we include the description of STP discussed above in our network model (*Equation 1* of the main text) by adding variables $x_{AB}$ and $u_{AB}$ for all projections $W_{AB}$, so that the synaptic current $W_{AB}r_B$ becomes $x_{AB}u_{AB}W_{AB}r_B$. Therefore, the network dynamics is given by

$$\begin{cases} \tau_E \frac{dr_E}{dt} & = -r_E + [x_{EE}\Omega_{EE}W_{EE}\,r_E - x_{EI}\Omega_{EI}W_{EI}\,r_I + I_{EX} - x_{0E}]_+ \\ \tau_I \frac{dr_I}{dt} & = -r_I + [x_{IE}\Omega_{IE}W_{IE}\,r_E - x_{II}\Omega_{II}W_{II}\,r_I + I_{IX} + \lambda L - x_{0I}]_+ \\ \frac{du_{AB}}{dt} & = -\frac{u_{AB}}{\tau_F} + U(1 - u_{AB})\,r, \\ \frac{dx_{AB}}{dt} & = \frac{1 - x_{AB}}{\tau_D} - \Omega_{AB}x_{AB}\,r, \\ \Omega_{AB} & = [u_{AB} + U(1 - u_{AB})] . \end{cases} \tag{A3}$$

Here $A, B \in [E, I]$ and, to simplify the analysis, we assumed threshold-linear single neuron transfer function.

## 2. Conditions for paradoxical response in networks with STP

In this section, we investigate conditions for paradoxical inhibitory response in networks with STP; the role of inhibition in stabilizing activity is discussed in the next section.

As discussed in *Tsodyks et al., 1997*; *Ozeki et al., 2009* and in *Figure 1—figure supplement 1*, paradoxical inhibitory response can be studied using the $r_E$ and $r_I$ nullclines of the model in the $r_E/r_I$ plane. The difference with those previous studies is that we have additional dynamical variables for synaptic strengths, and these should be set to their equilibrium values. Using $dx_{AB}/dt = 0$ and $du_{AB}/dt = 0$, we express $\Omega_{AB}$ and $x_{AB}$ as a function of $r_B$ (as done in *Equation A2*) and define

$$f_E(r_I) = x_{EI}(r_I)\Omega_{EI}(r_I)W_{EI}\,r_I, \quad f_I(r_I) = r_I + x_{II}(r_I)\Omega_{II}(r_I)W_{II}\,r_I . \tag{A4}$$

Since each term $x_{AB}(r_B)u_{AB}(r_B)W_{AB}r_B$ is a monotonically increasing function of the presynaptic rate $r_B$, $f_{E,I}$ can be inverted and, in the region of rates $r_{E,I} > 0$, we can write the excitatory and inhibitory nullclines as

$$\begin{cases} r_I^{excitatory}(r_E) & = f_E^{-1}[x_{EE}(r_E)u_{EE}(r_E)W_{EE}\,r_E - r_E + I_{EX} - x_{0E}] \\ r_I^{inhibitory}(r_E, L) & = f_I^{-1}[x_{IE}(r_E)u_{IE}(r_E)W_{IE}\,r_E + I_{IX} + \lambda L - x_{0I}] . \end{cases} \tag{A5}$$

Using the fact that $f_I^{-1}[.]$ is an increasing function of its argument, we obtain that the inhibitory nullcline $r_I^{inhibitory}(r_E, L)$ increases with $r_E$ and $L$. It follows that the response of the inhibitory population is paradoxical, that is the stationary value of $r_I$ decreases with $L$, only if the excitatory nullcline $r_I^{excitatory}(r_E)$ has positive slope, that is if

$$\frac{df_E^{-1}(r_E)}{dr_E} > 0 . \tag{A6}$$

*Equation A6* must be satisfied by any network of excitatory and inhibitory neurons with STP which shows paradoxical effect. In the next section, we show that this condition is violated in networks which are stable with fixed inhibition.

To conclude, we note that the condition of *Equation A6* is only necessary but not sufficient, and in order for the paradoxical response to emerge, one also needs $df_E^{-1}(r_E)/dr_E < df_I^{-1}(r_E)/dr_E$. Without STP, this condition is met any time the network is dynamically stable (*Ozeki et al., 2009*). With STP, proving an analogous result is more complicated, as it involves the stability of *Equation A3*, that is of a ten-dimensional dynamical system.

## Inhibition stabilization in networks with STP

In this section, we investigate the relation between paradoxical effect and inhibitory stabilization in networks with STP. As discussed above, plasticity can be included in all the synapses of the model (i.e. at all projections $W_{AB}$ in *Equation 1*) but only STP at recurrent excitatory synapses ($W_{EE}$) can influence the role of inhibition in the stabilization of dynamics. In fact, a network is said to be inhibition stabilized if instabilities emerge once inhibitory rates are

forced to be constant in time (*Tsodyks et al., 1997*; *Ozeki et al., 2009*). This condition is not affected by plasticity involving inhibitory neurons (both pre- and post- synaptically), therefore we can limit our investigation to STP in $W_{EE}$. In what follows, first we study analytically networks in which only one between STD or STF is present, then we investigate numerically networks in which both STD and STF are included simultaneously. In all these cases, we find that paradoxical inhibitory response implies that the network is unstable with fixed inhibition. Therefore, if a stable network shows paradoxical response, the excitatory network must be unstable on its own and has to be stabilized by inhibition. It follows that the conclusions derived in the main text about cortical networks remain valid also when STP is taken into account in the analysis.

## Effects of STD

With STD in $W_{EE}$ synapses, the network dynamics is given by

$$\begin{cases} \tau_E \frac{dr_E}{dt} = -r_E + [xW_{EE}r_E - W_{EI}r_I + I_{EX} - x_{0E}]_+ \\ \tau_I \frac{dr_I}{dt} = -r_I + [W_{IE}r_E - W_{II}r_I + I_{IX} + \lambda L - x_{0I}]_+ \\ \frac{dx}{dt} = \frac{1-x}{\tau_D} - xUr_E . \end{cases} \quad (A7)$$

The effect of STD on the stationary response can be seen by looking at the $r_E$ and $r_I$ nullclines of the system (*Appendix 1—figure 1A*) in the $r_E/r_I$ plane, setting $x$ to its steady-state value. Expressing $x$ as a function of $r_E$, we find

$$\begin{cases} r_I^{excitatory}(r_E) = \frac{1}{W_{EI}}\left[\left(\frac{W_{EE}}{1+\tau_D Ur_E} - 1\right)r_E + I_{EX} - x_{0E}\right], \\ r_I^{inhibitory}(r_E, L) = \frac{1}{1+W_{II}}[W_{IE}r_E + I_{IX} + \lambda L - x_{0I}] \end{cases} \quad (A8)$$

The above expression shows that STD decreases the self-excitation of excitatory cells as the excitatory rate increases. The stationary solutions of the *Equation A7* (here referred to as $x^*$, $r_E^*$, $r_I^*$) are the points at which the two nullclines of *Equation A8* cross. As discussed in general terms in the previous section, increasing $L$ in *Equation A8* moves the inhibitory nullcline upward and *Equation A6* provides a necessary condition for the paradoxical inhibitory response to emerge; this condition is met any time that

$$x^* > \frac{1}{\sqrt{W_{EE}}} . \quad (A9)$$

If *Equation A9* is not satisfied, the network response is not paradoxical.

The stability of the stationary solutions is found with a standard perturbation analysis around the stationary state. With frozen inhibition, we can write perturbation around the stationary solutions as $x = x^* + \delta_x$ and $r_E = r_E^* + \delta_E$. In the case of $r_E^* > 0$, the dynamics of these perturbations is given by

$$\begin{cases} \tau_E \frac{d\delta_E}{dt} = (-1 + x^*W_{EE})\delta_E + W_{EE}r_E^*\delta_x \\ \frac{d\delta_x}{dt} = -x^*U\delta_E - \left(\frac{1}{\tau_D} + Ur_E^*\right)\delta_x \end{cases} . \quad (A10)$$

*Equation A10* shows that, with fixed inhibition and fixed STD ($\delta_x = 0$), the excitatory network is unstable if

$$x^*W_{EE} > 1 \quad (A11)$$

which, since $0 < x < 1$, also implies that $W_{EE} > 1$. With fixed inhibition and dynamic STD, the dynamics is stable if

$$x^* < \min\left(\frac{1}{\sqrt{W_{EE}}}, \frac{1 + \sqrt{1 + 4\frac{\tau_E}{\tau_D}W_{EE}}}{2W_{EE}}\right) \quad (A12)$$

If **Equation A12** is verified, the network is stable without dynamic inhibition. Since this condition cannot be satisfied simultaneously with **Equation A9**, stabilization in a network with STD in $W_{EE}$ synapses which shows paradoxical effect has to be realized by inhibition.

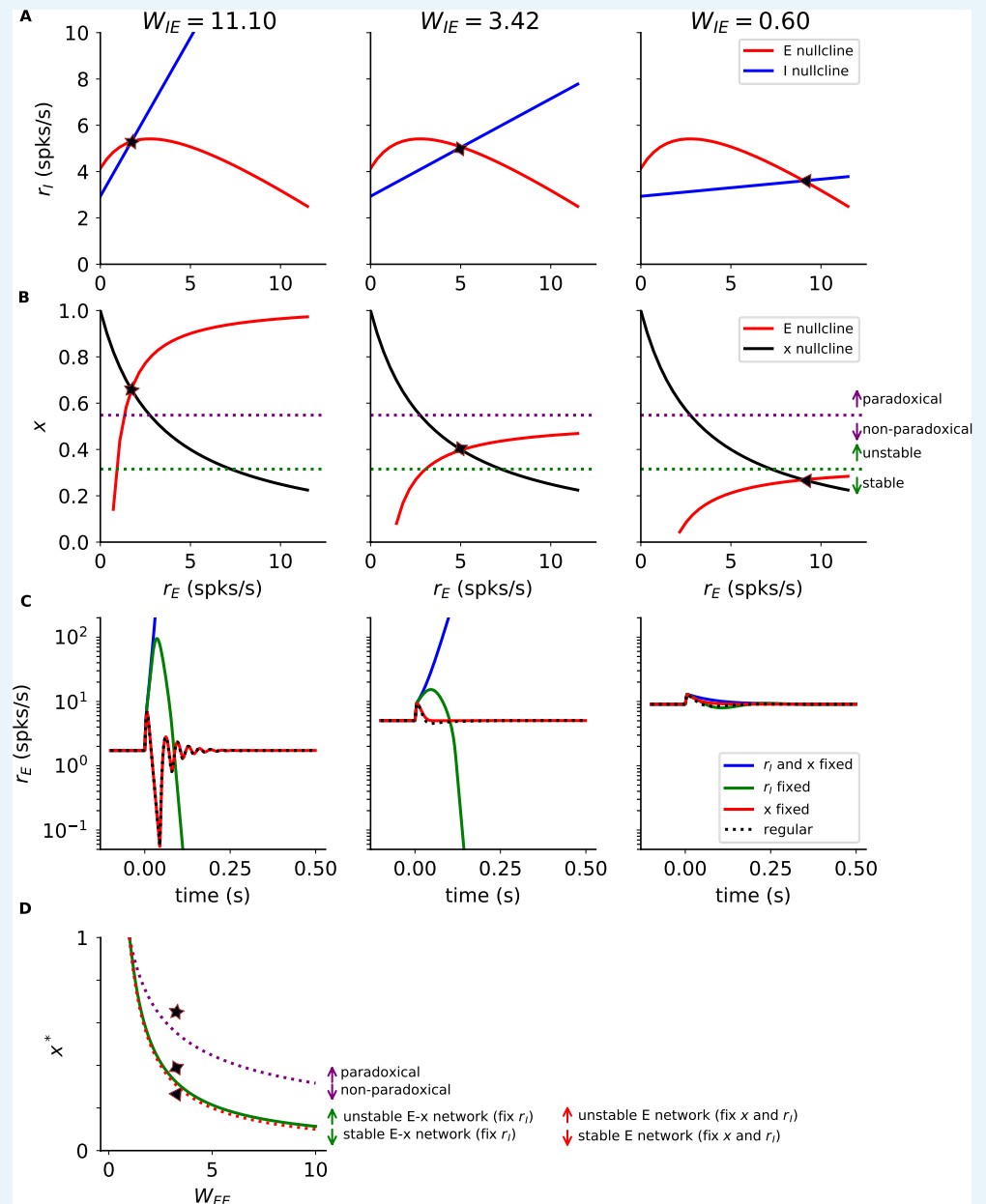

**Appendix 1—figure 1.** Network dynamics with short term depression. (**A**) Excitatory (red) and inhibitory (blue) nullclines of **Equation A7**. Different columns correspond to different values of $W_{IE}$ (from left to right, $W_{IE} = 11.10, 3.42, 0.60$); symbols correspond to the stationary solutions of the **Equation A7**. STD bends the excitatory nullcline downward, with a decrease that starts when $r_E \sim 1/U\tau_D$. Since increasing $L$ moves the inhibitory nullcline to the left, paradoxical inhibitory response emerges only if the nullclines meet in the region of positive slope of the excitatory nullcline. (**B**) As in (**A**) but for Excitatory (red) and x (black) nullclines of **Equation A7**. Dashed lines correspond to threshold for paradoxical response (purple, **Equation A9**) and stability of the excitatory population with fixed $x$ and $r_I$ (green, **Equation A12**). Note that paradoxical response can appear only if the network with fixed $r_I$ is unstable. Therefore, if a network is stabilized by STD, it cannot show paradoxical response. (**C**)

Dynamics of excitatory neurons in response to perturbations applied at time $t = 0$. Different colors correspond to different constraints on the dynamics. Different columns are as in panels A and B. (**D**) Values of $x^*$ as a function of $W_{EE}$ for which the network: has paradoxical response (**Equation A9**, purple), is stable with fixed $r_I$ (**Equation A12**, green), is stable with fixed $x$ and $r_I$ (**Equation A11**, red). Paradoxical inhibitory response emerges only if the excitatory subnetwork is unstable on its own ($W_{EE} > 1$) and with STD and fixed inhibition (**Equation A12**). Symbols correspond to the stationary solutions of panels A and B. Simulation parameters are: $W_{EE} = 3.33$, $W_{EI} = 1.77$, $W_{II} = 7.11$, $I_{EX} = 8.51$, $I_{IX} = 32.45$, $\lambda = 6.3$, $x_{0E} = 1.19$, $x_{0I} = 8.65$, $\tau_E = 7.8$ ms, $\tau_I = 34.3$ ms, $\tau_D = 500$ ms, $U = 0.6$.

## Effects of STF

In the presence of STF in $W_{EE}$ synapses, the network dynamics is given by

$$\begin{cases} \tau_E \frac{dr_E}{dt} = -r_E + [\Omega W_{EE}\ r_E - W_{EI}\ r_I + I_{EX} - x_{0E}]_+ \\ \tau_I \frac{dr_I}{dt} = -r_I + [W_{IE}\ r_E - W_{II}\ r_I + I_{IX} + \lambda L - x_{0I}]_+ \\ \frac{d\Omega}{dt} = -\Omega \frac{1 + U r_E \tau_F}{\tau_F} + U \frac{1 + r_E \tau_F}{\tau_F} \end{cases} \tag{A13}$$

Proceeding as in the case of STD, we find that inhibitory response is paradoxical, that is **Equation A6** is verified, if

$$W_{EE} > 1 \quad \text{and} \quad \Omega^* > 1 - \sqrt{1 - \frac{1 + U(W_{EE} - 1)}{W_{EE}}}. \tag{A14}$$

On the other hand, dynamics with fixed inhibition is stable if

$$\Omega^* < 1 - \sqrt{1 - \frac{1 + U(W_{EE} - 1)}{W_{EE}}} \tag{A15}$$

Therefore, as in the case of STD, stability with frozen inhibition is incompatible with paradoxical effect and paradoxical inhibitory response implies inhibitory stabilization.

## Analysis of networks with STF and STP in E→E synapses

To conclude our analysis, we investigate the relation between paradoxical inhibitory response and inhibitory stabilization in networks with both STF and STD in $W_{EE}$ synapses. In these networks, the dynamics is given by

$$\begin{cases} \tau_E \frac{dr_E}{dt} = -r_E + [x\Omega W_{EE}\ r_E - W_{EI} r_I + I_{EX} - x_{E0}]_+ \\ \tau_I \frac{dr_I}{dt} = -r_I + [W_{IE}\ r_E - W_{II} r_I + I_{IX} - x_{I0} + \lambda r_I]_+ \\ \frac{du}{dt} = -\frac{u}{\tau_F} + U(1 - u) r_E \\ \frac{dx}{dt} = \frac{1-x}{\tau_D} - \Omega x r_E, \quad \Omega = u + U(1 - u) \end{cases} \tag{A16}$$

The analytical approach we used to study networks with either STD or STF cannot be applied here, as finding the stationary solutions of **Equation A16** requires solving a system of equations with polynomials of degree higher than two in $r_E$. Therefore, to investigate the relation between paradoxical inhibitory response and inhibitory stabilization, we used a numerical approach.

We considered networks with randomly generated parameters and, for each realization, we analyzed numerically the response to external excitation of inhibitory neurons and the role of inhibition in stabilizing the dynamics. We considered networks with $\tau_E = \tau_I = 10$ ms, $\lambda = 1$, $x_{0E} = x_{0I} = 0$ and randomly generated $W_{AB}, I_{AX} \in \{0, 10\}$, $\tau_{F,D} \in \{0.01\text{s},\ 0.99\text{s}\}$ and $U \in \{0.01, 0.99\}$, using uniform distributions in the specified intervals. For each network, that is for each realization of the randomly-generated parameters, we simulated the dynamics induced by **Equation A16** for a time time $T = 10$ s (Euler method, integration time step $dt = 0.1$ ms) and measured the stationary values of $r_E$, $r_I$, $u$ and $x$. We assessed numerically that

each solution was stationary, and that rates were not oscillating or running away, by measuring the standard deviation of $r_E$ for $t>5$ s; we considered the dynamics stationary if, for $t>5$ s, std $(r_E)$/ mean $(r_E)<10^{-3}$. For each network with stationary dynamics, we ran another simulation ($T = 10$ s) and measured the response of inhibitory neurons to external stimulation. Specifically, we increased the value of $L$ from to 0.1 for $t \in \{5, 10\}$ s and considered the response to be paradoxical if $(r_I(L=0) - r_I(L=0.1))/(r_I(L=0) + r_I(L=0.1))>10^{-3}$, where $r_I(L=z)$ represents the rate measured at the end of the simulation period ($t = T = 10$ s) for $L = z$. To investigate the role of inhibition in stabilizing the dynamics, we ran yet another simulation ($T = 10$ s) in which, for time $t>5$ s, we clamped the value of inhibitory rates and applied a perturbation to excitatory neurons (increase in 0.05 spk/s of $r_E$ for $t>5.00$ s, $t<5.01$ s). We considered the network to be inhibition-stabilized if $|r_E(P = 0.05\,\mathrm{spk/s}) - r_E(P = 0\,\mathrm{spk/s})|/(r_E(P = 0.05\mathrm{spk/s}) + r_E(P = 0\mathrm{spk/s}))>10^{-3}$, where $r_E(P = z)$ represents the rate measured at the end of the simulation period ($t = T = 10$ s) with a perturbation of $z$ spk/s. Applying this approach to $N = 998845$ networks, we found that 98% (979227/998845) of them had stationary dynamics. Of these, 65.5% (639245/979227) were inhibition stabilized and 9% (88559/979227) had paradoxical inhibitory response. Of the inhibition stabilized networks, only 14% (88559/639245) showed paradoxical inhibitory response. This is consistent with our analytical results in networks with either STD or STP, where we found that inhibitory stabilization does not imply paradoxical inhibitory response. Of the networks with paradoxical response, consistent with what we found in networks with either STD or STP, 100% (88559/88559) were inhibition stabilized.

Our numerical results suggest that, also in networks in which STD and STF coexist in $W_{EE}$ synapses, paradoxical inhibitory response implies inhibitory stabilization.

## Appendix 2

### Response analysis with networks of leaky integrate and fire neurons

In the main text, using rate models without input noise, we have shown that experimental data recorded in V1 are consistent with the network operating in the inhibition stabilized regime, with mean excitatory and inhibitory inputs larger than threshold and total input of order threshold. In this section we show that similar results are obtained using a rate model which features input noise and a more biologically accurate transfer function. Furthermore, this analysis shows that inferred parameters agree with previously measured network properties, and that data are consistent with the loose balance regime (**Ahmadian and Miller, 2019**) and noise driven firing.

### Model definition

We use a network of excitatory and inhibitory neurons analogous to the one described in the main text, but with a single neuron transfer function matching that of leaky integrate and fire models. In the network, each neuron in population $A = [E, I]$ receives a Gaussian distributed input, of mean $\mu_A$ and variance $\sigma_A^2$, which is produced by recurrent and feedforward interactions. As shown in **Brunel, 2000**, population firing rates are found by solving:

$$\begin{cases} r_E = \left[ \tau_{rp} + \tau_E \sqrt{\pi} \int_{u_{min,E}}^{u_{max,E}} e^{u^2} \left(1 + \mathrm{erf}(u)\right) du \right]^{-1}, \\ r_I = \left[ \tau_{rp} + \tau_I \sqrt{\pi} \int_{u_{min,I}}^{u_{max,I}} e^{u^2} \left(1 + \mathrm{erf}(u)\right) du \right]^{-1}; \end{cases} \tag{B1}$$

where $\tau_{rp}$ is the single neuron refractory period, while $u_{max,A}$ and $u_{min,A}$ are the distance from spiking threshold $\theta$ and reset $V_r$ of the mean input $\mu_A$ measured in units of input noise $\sigma_A$, that is

$$u_{max,A} = \frac{\theta - \mu_A}{\sigma_A}, \quad u_{min,A} = \frac{V_r - \mu_A}{\sigma_A}. \tag{B2}$$

In the model, using the same notation as in the linear model of the main text, means are given by

$$\begin{aligned} \mu_E &= I_{EX} + W_{EE} r_E - W_{EI} r_I, \\ \mu_I &= I_{IX} + W_{IE} r_E - W_{II} r_I + \lambda L; \end{aligned} \tag{B3}$$

we assume noise amplitude to be fixed to $\sigma_E = \sigma_I = 5\mathrm{mV}$ (**Haider et al., 2013**). Other model parameters are: $\tau_{rp} = 2\,\mathrm{ms}$, $\tau_E = 20\,\mathrm{ms}$, $\tau_I = 10\,\mathrm{ms}$, $\theta = 20\,\mathrm{mV}$, $V_r = 10\,\mathrm{mV}$. This model has a total of 9 free parameters (four connectivity parameters $W_{AB}$, two feed-forward inputs $I_{AX}$, one laser efficacy $\lambda$, two blockers efficacy $\epsilon_A$) which will be inferred from the data.

### Fitting procedure

Evaluating rates from **Equation B1** is computationally expensive; to speed up the exploration of the parameter space, we fitted the model directly to the average population response as follows.

We discretized laser intensities into 101 bins centered around laser values $L_i$ and equally spaced in the interval $[0, L_{max}]$ ($L_{max}$ is the maximum $L/L_0$ common to all recording days, we used $L_{max} = 15$ and checked that results were unchanged for $L_{max} = 10 - 14$). The error in the model description was computed as

$$Err = \sum_{A\in[E,I],phase\in[1,2,3]} Err_{A,phase},$$

$$Err_{A,phase} = \frac{1}{\text{number of points with } L_i \leq L_{in}} \sum_{L_i \leq L_{in}} \left[\frac{data(L_i) - model(L_i)}{\max(sem(L_i),1)}\right]^2 \qquad \text{(B4)}$$

$$+ \text{analogous expression for } L_{in} < L_i < L_{max},$$

where $model(L_i)$ is the model prediction obtained from **Equation B1** for $L = L_i$, while $data(L_i)$ ($sem(L_i)$) is the mean rate (standard error of the mean); this was computed averaging responses from all cells to stimuli of intensity $L/L_0$ within the bin centered around $L_i$. The parameter $L_{in}$ divides data into two groups, $L_i < L_{in}$ and $L_{in} < L_i$, whose contributions to the error are weighed by the corresponding number of data points; this split was used to prevent the large laser responses of inhibitory cells from dominating the error (we used $L_{in} = 1$ and checked that results were robust for $L_{in} = 2, 3$). The minimum bound on the standard error of the mean ($\max(sem(L_i), 1)$) prevents regions with low $sem(L_i)$ (specifically, the large laser intensity response of excitatory cells) from dominating the error.

Given the average neural responses of different populations and phases, optimal model parameters were determined by minimizing **Equation B4** as described in the methods section of the main text. Finally, to better estimate the excitatory rate at large laser intensity, we included in our analysis only cells that were classified in the same way (inhibitory or excitatory) from the classification done with blockers and with response at high laser intensity. This filtering reduces the number of cells used in the analysis, from 167 (111 E+ 56 I) to 119 (90 E + 29 I), and decreases the excitatory rate measured at large laser intensity (**Appendix 2—figure 1A**).

## Results of the analysis

The best model description of the data, obtained minimizing **Equation B4**, is shown in **Appendix 2—figure 1A**. The corresponding model parameters are:

$$W_{EE} = 2.65\text{mVs}, W_{EI} = 2.28\text{mVs}, W_{IE} = 3.03\text{mVs}, W_{II} = 2.50\text{mVs},$$
$$I_{EX} = 20.19\text{mV}, I_{IX} = 19.49\text{mV}, \lambda = 3.38\text{mV}, \epsilon_E = 0.55, \epsilon_I = 0.32. \qquad \text{(B5)}$$

As in the main text, we used a bootstrap approach to estimate the precision with which model parameters can be inferred from the data; results are shown in **Appendix 2—figure 1B**. The analysis shows that distribution of inferred parameters are localized around the optimal solution described above. Moreover, the parameters of the model can be map to known biophysical quantities by noticing that $W_{EE} = \tau_E K_{EE} J_{EE}$ (**Brunel, 2000**), where $K_{EE}$ and $J_{EE}$ are the average number of recurrent excitatory projection and their efficacy (analogous expression holds for the other elements of the matrix $W$). Assuming $\tau_E = 20$ ms and $K_{EE} \approx 10^2 - 10^3$, we obtain an estimate of $J_{EE} \approx 0.1 - 1$ mV, which is consistent with direct biophysical measurements (**Holmgren et al., 2003**).

We computed the inputs into excitatory cells; contributions coming from recurrent inhibition ($-W_{EI}r_I$), recurrent excitation ($W_{EE}r_E$), and feed-forward excitation ($I_{EX}$) are shown in **Appendix 2—figure 1C**. The total input is found to be below threshold, at a distance of order one in unit of input noise. This result is consistent with the balanced state model (**Amit and Brunel, 1997**; **van Vreeswijk and Sompolinsky, 1996**; **van Vreeswijk and Sompolinsky, 1998**), where firing is driven by input noise. However, unlike what is expected by the balanced state model, the different components of the input (e.g. feed-forward excitation) are of order threshold, consistent with the loose balance regime recently suggested to underlie cortical dynamics (**Ahmadian and Miller, 2019**).

In the model, the strength of self-excitation of the excitatory population is obtained from **Equation B1** as

$$\mathcal{W}_{EE} = \frac{d}{dr_E}\left[\tau_{rp} + \tau_E\sqrt{\pi}\int_{u_{min,E}}^{u_{max,E}} e^{u^2}(1 + \text{erf}(u))\,du\right]^{-1}, \qquad \text{(B6)}$$

where the integral is evaluated at the inferred baseline activity. The quantity $\mathcal{W}_{EE}$ is the generalization of the parameter $W_{EE}$ analyzed in the main text, which takes into account the amplification of coupling strength due to transfer function nonlinearities. Values of $\mathcal{W}_{EE}$ inferred with data bootstrap are shown in *Appendix 2—figure 1D*. Results are consistent with $\mathcal{W}_{EE}$ larger than one, indicating that indeed the excitatory subnetwork is unstable without inhibition, and in the range 5–15, consistent with the result obtained in the main text using a linear model.

To summarize, the analysis derived in this section shows that results obtained in the main text are preserved when a more biologically accurate, spiking, model is used to describe the data. Moreover, this approach also shows that inferred parameters are consistent with known biophysical properties of cortical networks.

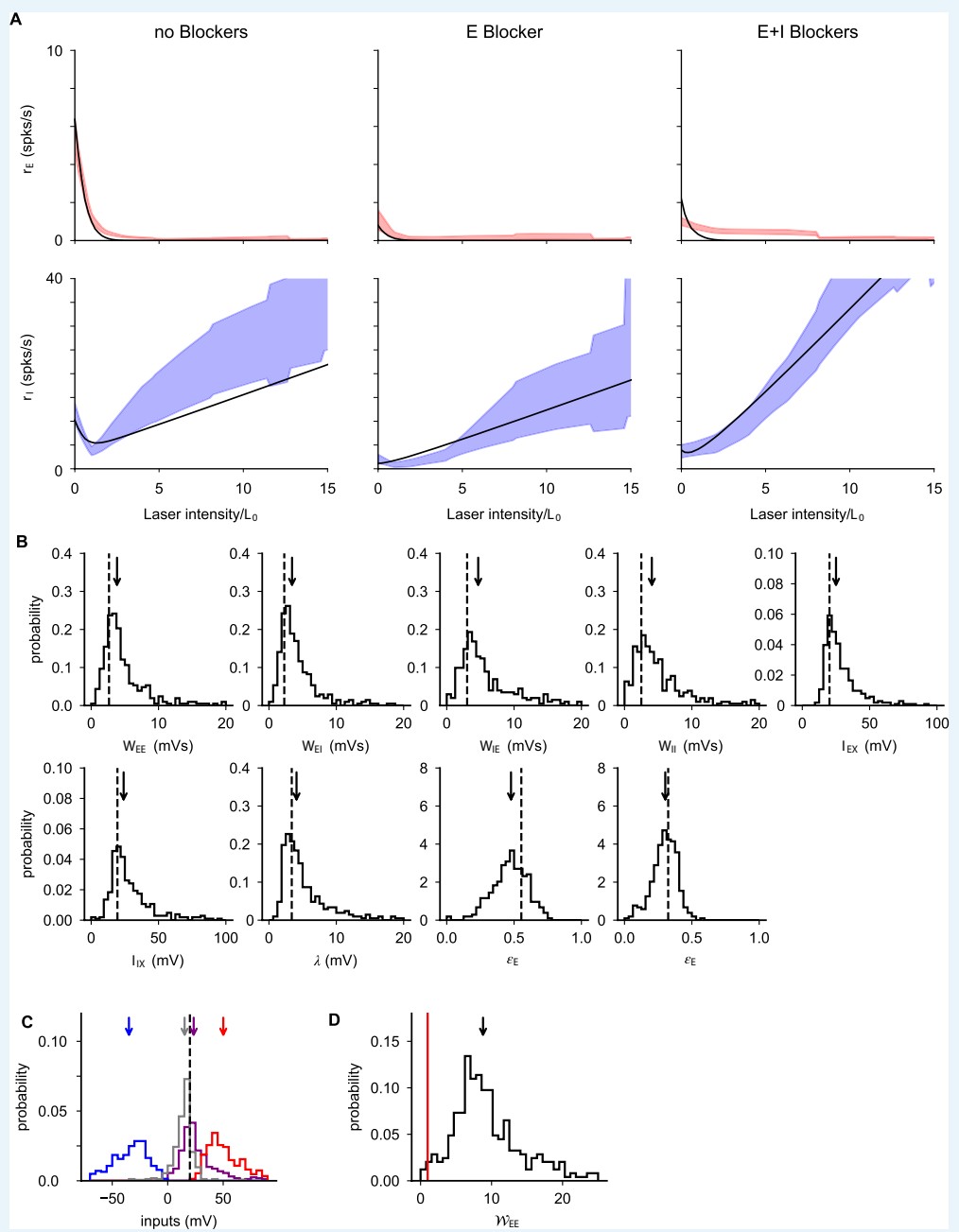

**Appendix 2—figure 1.** Population response in the three phases of the experiment. (**A**) Optimal fit of population response given by *Equation B1*. (**B**) Distributions (lines) and medians (arrows)

of inferred parameters obtained with data bootstrap; dashed lines are optimal parameters reported in the text. (C) Distribution (line) and median (arrow) of self excitation of excitatory cells computed from data using *Equation B6*. The inferred values are distributed above the instability point $W_{EE} = 1$ (red line). (D) Distributions (lines) and medians (arrows) of recurrent excitation ($W_{EE}r_E$, purple), recurrent inhibition ($-W_{EI}r_I$, blue), feed-forward+recurrent excitation ($W_{EE}r_E + I_{EX}$, red), and total input ($-W_{EI}r_I + W_{EE}r_E + I_{EX}$, gray) to excitatory cells. As discussed in the text, inputs are of order threshold (dashed line); data are consistent with fluctuation driven firing (*Amit and Brunel, 1997*; *van Vreeswijk and Sompolinsky, 1996*; *van Vreeswijk and Sompolinsky, 1998*) and loose balance regime (*Ahmadian and Miller, 2019*).

