## [Decision Letter]

**Acceptance summary:**

This study thoroughly addresses a long-standing question in cortical neurophysiology, namely the extent to which cortex resides in an inhibition stabilised regime. Using careful experiments and mathematical modelling the authors reveal several hallmarks of this regime in multiple cortical areas, including so-called paradoxical inhibition of inhibitory interneuron firing rate, in both awake and anaesthetised states. Together, these results provide further compelling evidence of the role of inhibition in stabilising neural activity.

**Decision letter after peer review:**

Thank you for submitting your article "Inhibition stabilization is a widespread property of cortical networks" for consideration by *eLife*. Your article has been reviewed by three peer reviewers, one of whom is a member of our Board of Reviewing Editors, and the evaluation has been overseen by John Huguenard as the Senior Editor. The following individual involved in review of your submission has agreed to reveal their identity: Hillel Adesnik (Reviewer #3).

The reviewers have discussed the reviews with one another and the Reviewing Editor has drafted this decision to help you prepare a revised submission.

This work address whether excitatory cortical activity is stabilised by rapid inhibitory feedback across cortical areas. One important prediction of such a model is so-called paradoxical suppression of inhibitory neurons in response to inhibitory stimulation. This prediction is validated experimentally and analysed with mathematical modelling.

The reviewers broadly agree that the study is thorough, well executed and well presented. Several issues were raised, including the need to show/quantify more raw data and population data. There are also some potential experimental confounds, explained in detail below, that should be addressed either by appropriately adjusting conclusions or if possible, by reanalysis or modelling. Finally, for the wide readership *eLife* caters for, the authors should make more effort to explain the importance and relevant functional consequences of the ISN regime.

Reviewer #1:

This is a well written paper without any major deficiencies. The authors address the question whether the activity of excitatory neurons is stabilised by rapid inhibitory feedback, and how common this feature is across cortical areas. The hypothesis that such inhibition-stabilised networks (ISN) would display the so-called 'paradoxical suppression' is intuitive and nicely backed up by a Wilson-Cowan model. The data does indeed show the paradoxical suppression, supporting the ISN hypothesis nicely.

The authors also address another possible explanation of the paradoxical suppression, namely the suppression of inhibition-to-inhibition coupling, and explain why this is less convincing, again supported by data. They also go into some depth on the strength of the coupling, which seems to be moderate, rather than very strong.

It would help the wider readership if the authors could go into more depth on why this question is important. It seems self-evident that cortical networks are stabilised by inhibition: few other mechanisms are rapid enough and moderate loss of inhibition generically leads to seizures.

For example, what are the computational benefits of an ISN? This is only briefly touched upon in the end of the Discussion section at the moment, and could be elaborated upon in the Introduction more.

Reviewer #2:

The paper of Sanzeni et al., demonstrates that the firing rate of inhibitory cells expressing ChR2 is suppressed at low light intensity but as intensity increases firing rate return and then becomes even higher. This is a paradoxical behavior since we expect that firing rate will increase monotonically with light intensity. The results are in a good agreement with predictions made by inhibition-stabilized network (ISN) models. Such behavior was found in superficial and deep layers of different cortical areas of awake mice and also in anesthetized mice.

The methods of the experiments are straightforward and were conducted with high standards. The presentations of the results are clear and nice. For example, the classification of cells into inhibitory and excitatory cells based on different methods is convincing. The virus vs. transgenic experiments are also clear and important. The comparisons of the effect of light before and after addition of synaptic blockers are important and help in clarification of the experiments. Finally, the demonstration of the effect in deep layers and under anesthesia are crucial. Thus, the experimental work, regardless the model, is interesting and important.

However, this study is not novel enough as similar conclusions were shown in previous studies and in particular in the 2018 study of Moore et al., from Wehr lab which presented a paradoxical increase of PV firing rate when these cells were optogenetically suppressed (the authors cite this study). Since Moore's paper includes a substantial modeling of ISN network, I am not sure if this under review study introduces novel concepts (the major difference is that in this study inhibitory cells were activated and in the 2018 paper the cells were inactivated).

As an experimentalist I have some concerns related to some experimental aspects. Unlike the previous studies, no direct measurements of excitatory and inhibitory currents were made in this study and thus my enthusiasm was reduced. Yet, the study can benefit from pharmacology, which can be better trusted compared to optogenetics.

Essential revisions:

1) Unlike the light-inactivation of inhibitory cells using Arch, illumination of the cortex of VGAT-ChR2 mice can directly cause synaptic release from the terminals, regardless the firing of the cells. I mention this issue although I don't think that it is fully understood. In other words, it is unclear if the firing of the cells directly reports the effect of light in the network (which could be better assessed using patch recordings). Ideally this could be addressed by limiting the expression of ChR2 to cell bodies. Since none of my proposed experiments are realistic for a revision, I propose to discuss this issue in detail or even try to model it.

2) Pharmacology: the effect of GABAr blockers should be tested before blocking excitatory transmission. Adding such blockers at low concentration to partially reduce the effect of inhibition perhaps will result with a similar (but opposite) effect to the optogenetic and perfusion of 3 or 4 concentrations in an increasing manner may work.

3) It is not clear if and where the population distribution of the paradoxical reduction in the firing rate of inhibitory cells is presented in the paper. It is possible that firing rate of excitatory cells is suppressed due a small fraction of inhibitory cells for which firing rate increases monotonically with light intensity.

4) As far as I understand the model is not a conductance based model. I think that providing predictions for currents can greatly help in illustrating the mechanisms (simply feed the firing rates of the cells into a single cell model).

5) The roles of short-term synaptic plasticity were not discussed or checked both in the model and in the experiments. Both depression and facilitation can contribute. Different levels of depression/facilitation of inhibitory synapses due to their activation may contribute to the effect. Without modeling them it is hard to predict their contributions.

6) Rather than changing light intensity the authors should consider increasing the surface area that is illuminated. I think that the model will predict that this will result with similar effect to that for increasing light intensity. I believe that this approach is better.

Reviewer #3:

Sanzeni et al., present a compelling new data set supporting the notion that neocortical areas operate in the inhibition stabilized regime in which recurrent inhibition is needed to balance recurrent excitation to stabilize the network. This had been theoretically proposed more than 20 years ago by Tsodyks and colleagues, and experimental data to support it was originally put forward by Ferster, Miller and colleagues in 2009 using whole cell recording in anesthetized cats. Subsequently, additional whole cell recordings in awake mice in A1 and V1 supported this model. What has been lacking was a direct test of a core prediction of the ISN model: that direct excitation of inhibitory neurons (I cells) would paradoxically lead to a net suppression of I cells at the steady state (followed by a transient increase in I cell activity). It should be clear this result was indirectly shown by measuring synaptic inhibition in the Ferster and Miller paper. The current study elegantly and rigorously used direct optogenetic stimulation of I cells in transgenic mice to show that 'paradoxical' prediction is borne out in mouse V1. The authors then go on to repeat this core result in S1 and M1 and in deeper cortical layers of these areas. Overall, I find the study well performed and the arguments sound. I previously already subscribed to this view based on the existing data, but the data in this study really is the nail in the coffin. For this reason, this study is timely and warrants publication, with *eLife* a suitable journal.

I do have some questions for the authors to address (both conceptual and methodological). While I agree with almost all of the authors conclusions, I would like them to address a few points in which I will act as a devil's advocate. I'd like them to conceptually, experimentally, or theoretically rule out alternative explanations for the data, however less likely they might seem than the ISN model.

1) Could it be that a very small subset of PV+ interneurons are extremely photo-excitable (perhaps a subtype that suppresses all other cell types across all layers in cortex) and that these cells are in fact responsible for the suppression of most other I cells? These cells might be so rare (inhibitory 'hub' cells) that they would be very hard to catch with extracellular recording (or they might have small spikes that are below the noise threshold). Perhaps they have a slightly delayed response to optogenetic stimulation which explains the transient increase in I cell activity prior to its suppression?

2) Could a pure feed-forward model explain their results coupled with the possibility that L4 interneurons could be more photo-excitable than L2/3 interneurons, despite being deeper in the brain? In this scenario, the 'paradoxical' I cell suppression in L2/3 could be entirely explained by loss of feedforward excitation from L4. It seems like in the data set some I cells didn't show paradoxical effects (and most of the authors data comes from above L4), so maybe this is the case?

3) In any case, the authors should show much more raw data (rather than just average data) in the main figures. I agree it's the average that matters to test the ISN model, but it would be helpful to the reader to see the distribution across units for the various effects: transient activation, shape of 'paradoxical response' etc. Some of this is in the supplement, but not much.

---

## [Author Response]

This work address whether excitatory cortical activity is stabilised by rapid inhibitory feedback across cortical areas. One important prediction of such a model is so-called paradoxical suppression of inhibitory neurons in response to inhibitory stimulation. This prediction is validated experimentally and analysed with mathematical modelling.The reviewers broadly agree that the study is thorough, well executed and well presented. Several issues were raised, including the need to show/quantify more raw data and population data. There are also some potential experimental confounds, explained in detail below, that should be addressed either by appropriately adjusting conclusions or if possible, by reanalysis or modelling. Finally, for the wide readership eLife caters for, the authors should make more effort to explain the importance and relevant functional consequences of the ISN regime.Reviewer #1:This is a well written paper without any major deficiencies. The authors address the question whether the activity of excitatory neurons is stabilised by rapid inhibitory feedback, and how common this feature is across cortical areas. The hypothesis that such inhibition-stabilised networks (ISN) would display the so-called 'paradoxical suppression' is intuitive and nicely backed up by a Wilson-Cowan model. The data does indeed show the paradoxical suppression, supporting the ISN hypothesis nicely.

Thank you for the positive comments.

The authors also address another possible explanation of the paradoxical suppression, namely the suppression of inhibition-to-inhibition coupling, and explain why this is less convincing, again supported by data. They also go into some depth on the strength of the coupling, which seems to be moderate, rather than very strong.

Yes, our results support moderate, not very strong, recurrent connectivity. We have revised the Discussion section to highlight this point (see also Ahmadian and Miller, 2019).

It would help the wider readership if the authors could go into more depth on why this question is important. It seems self-evident that cortical networks are stabilised by inhibition: few other mechanisms are rapid enough and moderate loss of inhibition generically leads to seizures.

We agree with the reviewer that the presence of epileptic seizures upon blockade of inhibition seems at first sight to imply that cortex is inhibition-stabilized. However, this does not rule out that cortex is inhibition-stabilized in some states, but not others – in particular, it could be inhibition-stabilized in the presence of sensory inputs, but not during spontaneous activity. In fact, to the best of our knowledge, while there was already strong evidence for inhibition stabilization during sensory processing, the question of whether cortex is inhibition stabilized at baseline remained an open question until our study. We now make this point more clearly in the Introduction.

For example, what are the computational benefits of an ISN? This is only briefly touched upon in the end of the Discussion section at the moment, and could be elaborated upon in the Introduction more.

We have added to the first paragraph of the Introduction to discuss the computational benefits of ISNs, including now citing work on ‘balanced amplification’ (recent sensory cortex data suggests that some input patterns may be amplified; e.g. Marshel et al., 2019; see also Sanzeni and Histed, 2020 for brief discussion), and work on how strongly coupled recurrent networks can generate complex spatiotemporal patterns of activity (e.g. Hennequin et al., 2014).

Thank you for these suggestions; we think the manuscript has been greatly improved.

Reviewer #2:The paper of Sanzeni et al., demonstrates that the firing rate of inhibitory cells expressing ChR2 is suppressed at low light intensity but as intensity increases firing rate return and then becomes even higher. This is a paradoxical behavior since we expect that firing rate will increase monotonically with light intensity. The results are in a good agreement with predictions made by inhibition-stabilized network (ISN) models. Such behavior was found in superficial and deep layers of different cortical areas of awake mice and also in anesthetized mice.The methods of the experiments are straightforward and were conducted with high standards. The presentations of the results are clear and nice. For example, the classification of cells into inhibitory and excitatory cells based on different methods is convincing. The virus vs. transgenic experiments are also clear and important. The comparisons of the effect of light before and after addition of synaptic blockers are important and help in clarification of the experiments. Finally, the demonstration of the effect in deep layers and under anesthesia are crucial. Thus, the experimental work, regardless the model, is interesting and important.

Thank you for the positive comments and thorough review.

However, this study is not novel enough as similar conclusions were shown in previous studies and in particular in the 2018 study of Moore et al., from Wehr lab which presented a paradoxical increase of PV firing rate when these cells were optogenetically suppressed (the authors cite this study). Since Moore's paper includes a substantial modeling of ISN network, I am not sure if this under review study introduces novel concepts (the major difference is that in this study inhibitory cells were activated and in the 2018 paper the cells were inactivated).

While it is true that ISN effects have been seen before (cat V1, Ozeki et al., 2009; mouse A1, Kato et al., 2017), there has been an open question of whether cortical networks always operate as ISNs. Theoretical work has shown that networks can transition out of an ISN regime as network activity goes down, and theoreticians have actively speculated that as sensory stimuli grow in intensity, even cat V1 may transition from a non-ISN to an ISN state (for review, see Ahmadian and Miller, 2019). Further, even the question of whether the ISN regime generalizes outside cat V1 was questioned (see discussion in Rubin et al., 2015, and discussion of open ISN questions in the mouse in Litwin-Kumar et al., 2016). In fact, a recent paper from David Hansel and Nuo Li’s group (Mahrach et al., 2020), argues that PV cell stimulation as performed by Kato et al., etc, does not imply ISN operation.

Our present work shows clear evidence for the ISN state, relying on experimental stimulation of all classes of inhibitory neurons together.

The Moore et al. work is interesting, as they find paradoxical changes in firing rates in mouse A1, but that work neither proves nor disproves whether mouse auditory L2/3 is inhibition stabilized. Suppressing only L2/3 PV neurons, they find Arch-expressing cells decrease their firing rates (as expected for Arch-induced suppression), and Arch-negative cells increase their firing rate (as one might expect from disinhibition due to Arch suppression of other I cells). These observations do not provide evidence for an ISN interpretation, because, subject to some assumptions, in an ISN the cells stimulated by the opsin will produce paradoxical effects. In fact, the authors argue for a non-recurrent explanation. They find that a feedforward disinhibitory circuit (called a cascaded feedforward network in the paper), where L4 inhibitory cells disinhibit L2/3 inhibitory cells, explains their data. This feedforward disinhibition can be implemented in weakly-coupled networks, and does not require the network to be stabilized by inhibition.

There are several differences between the Moore paper and ours. First, they used viral expression of opsin, which, as we show, can change how paradoxical effects manifest (also see Sadeh et al., 2017, for more theory work on this point). Second, we showed, using synaptic blockers, that cells which are stimulated by the laser respond paradoxically, and we provide several lines of evidence (dynamics, two-phase inhibitory responses, self-consistent model, etc; outlined in the Discussion section) for the ISN mechanism, and against this effect being produced by disinhibitory circuits. Our result, showing strong recurrent coupling, combined with the Moore result, suggests that it will be interesting for future work to examine both across and within brain areas whether these two mechanisms might be used for different types of computation.

We have added text to better explain the above issues. In the Introduction we have revised the discussion of the non-ISN to ISN transition topic, and in both the Introduction and Discussion section we have revised for clarity the discussion of the Moore result.

As an experimentalist I have some concerns related to some experimental aspects. Unlike the previous studies, no direct measurements of excitatory and inhibitory currents were made in this study and thus my enthusiasm was reduced. Yet, the study can benefit from pharmacology, which can be better trusted compared to optogenetics.

Yes. We would add that the pharmacology work made possible the modeling (Figure 4) results, which would have been unobtainable without the extra pharmacological measurements.

Essential revisions:1) Unlike the light-inactivation of inhibitory cells using Arch, illumination of the cortex of VGAT-ChR2 mice can directly cause synaptic release from the terminals, regardless the firing of the cells. I mention this issue although I don't think that it is fully understood. In other words, it is unclear if the firing of the cells directly reports the effect of light in the network (which could be better assessed using patch recordings). Ideally this could be addressed by limiting the expression of ChR2 to cell bodies. Since none of my proposed experiments are realistic for a revision, I propose to discuss this issue in details or even try to model it.

Thanks for raising this point. For certain potential effects, such as those that depend on stimulating only the cells that are being measured, optogenetically-induced synaptic release from terminals (Zhao et al., 2011; Babl et al., 2019) might well change the interpretation of experiments. However, for the network effects produced in an ISN, the paradoxical effect is generated by I cells affecting the population of E cells, which then withdraw input from the population of I cells. Since ISN phenomena are population effects (dependent on I cells’ synaptic targets being affected, without specifically measuring each cell that was stimulated), whether target neurons are affected by direct synaptic release or by eliciting spikes at inhibitory neurons’ somata should not change the conclusions. Specifically, for both induced spiking and induced synaptic release, if the network is an ISN we expect firing rates to decrease with I stimulation (showing paradoxical suppression), as we found. Also, our pharmacology data supports the idea that many cells were induced to fire directly by stimulation. When we added E and I blockers, we still found that many cells fired in response to optogenetic stimulation. We have added to the Discussion section to discuss this.

2) Pharmacology: the effect of GABAr blockers should be tested before blocking excitatory transmission. Adding such blockers at low concentration to partially reduce the effect of inhibition perhaps will result with a similar (but opposite) effect to the optogenetic and perfusion of 3 or 4 concentrations in an increasing manner may work.

Yes, we also tried to verify the ISN prediction in this way. Adding GABA blockers in an ISN could in principle lead to a paradoxical effect of inhibitory rates increasing as excitatory rates increase. To look for this, we did indeed do pilot experiments in which we added GABA blockers first. While this did raise spontaneous excitatory firing rates, it was difficult to find a useful concentration that did not lead to epileptiform activity, so we settled on the sequence used in Figure 4. Due to this difficulty in precise control, we were not able to determine whether changes in inhibitory firing rates via pharmacological blockade also led to paradoxical effects. What made such measurements possible with optogenetics was the ability to more precisely control stimulation intensity and stimulation timing.

3) It is not clear if and where the population distribution of the paradoxical reduction in the firing rate of inhibitory cells is presented in the paper. It is possible that firing rate of excitatory cells is suppressed due a small fraction of inhibitory cells for which firing rate increases monotonically with light intensity.

We have added population data to the paper. We have plotted responses for every V1 inhibitory cell in the supplement (new Figure 2—figure supplement 7). We had previously shown the distribution across all cells of the paradoxical effect (initial slope, e.g. Figure 2JK, 5D,H, 6KL etc.), plotted as raw and normalized responses, and we have chosen to retain these. We have also added a set of raw example cells to Figure 2 (new panel I).

Also, the reviewer mentions the possibility that rare, non-paradoxical inhibitory cells could suppress all other I cells. Our data make this possibility unlikely. First, the observed dynamics (initial increase in inhibition followed by suppression, Figure 3) is inconsistent with this scenario. Second, we observed that suppressed inhibitory cells respond in a non-paradoxical way for large laser intensities, which is also inconsistent with the rare-I-cell scenario. If a small number of I cells control the effect, I cells should show similar responses at low and high stimulation intensity (majority: suppressed by the laser, rare: excited), not for I cells to transition from suppression to excitation in the way we observed. (Supporting this, we found that in transgenic animals, almost all I cells show paradoxical effects; e.g. see initial response slopes in Figure 2JK, new supp. figure.) Moreover, the transition (from paradoxical to non-paradoxical response with increasing stimulation) happens at the point at which E cells become suppressed, as predicted by ISN models. We have added a paragraph to make these points in the Discussion section.

4) As far as I understand the model is not a conductance based model. I think that providing predictions for currents can greatly help in illustrating the mechanisms (simply feed the firing rates of the cells into a single cell model).

We have added text to the Discussion section to say that ISN models make similar predictions for paradoxical effects in recorded intracellular currents and in the firing rates we recorded, and to connect this work to the past intracellular work by Kato et al., Moore et al., etc.

5) The roles of short-term synaptic plasticity were not discussed or checked both in the model and in the experiments. Both depression and facilitation can contribute. Different levels of depression/facilitation of inhibitory synapses due to their activation may contribute to the effect. Without modeling them it is hard to predict their contributions.

This is a very good question. To our knowledge, the way short-term plasticity impacts the relationship between paradoxical suppression and ISN operation has not been studied (surprisingly). Therefore, we explored this analytically in detail. We have added an appendix (Appendix A) to the manuscript with a figure. Our results show that when a paradoxical inhibitory response is observed in a network with synaptic plasticity, that implies the excitatory network is unstable and thus that the network is an ISN. We now explain these results in the main text (subsection “Mouse primary visual cortex is inhibition stabilized”).

6) Rather than changing light intensity the authors should consider increasing the surface area that is illuminated. I think that the model will predict that this will result with similar effect to that for increasing light intensity. I believe that this approach is better.

Illumination area and stimulation strength are partly related and partly orthogonal. Stimulating small numbers of inhibitory cells (as can happen with small illumination areas) is similar to stimulating a small fraction of inhibitory cells in a network. Therefore, for small illumination areas, paradoxical effects and ISN operation are no longer related (similar to what we found with inhibitory viral expression.), Sadeh et al., (2017) characterized in a model the impact of changing illumination area, and confirmed that for small areas (100 µm with their parameters), the network no longer shows paradoxical effects. In a sense, then, increasing area does produce changes in inhibitory response sign, as changing laser power does, though the mechanisms are not the same: increasing laser power shifts the network to a non-ISN state while decreasing illumination area leaves the network in the ISN state but hides the paradoxical effect. Thanks for this comment; to clarify this for readers we have added text to the Discussion section to explain this effect of area, citing Sadeh et al., there.

Reviewer #3:Sanzeni et al., present a compelling new data set supporting the notion that neocortical areas operate in the inhibition stabilized regime in which recurrent inhibition is needed to balance recurrent excitation to stabilize the network. This had been theoretically proposed more than 20 years ago by Tsodyks and colleagues, and experimental data to support it was originally put forward by Ferster, Miller and colleagues in 2009 using whole cell recording in anesthetized cats. Subsequently, additional whole cell recordings in awake mice in A1 and V1 supported this model. What has been lacking was a direct test of a core prediction of the ISN model: that direct excitation of inhibitory neurons (I cells) would paradoxically lead to a net suppression of I cells at the steady state (followed by a transient increase in I cell activity). It should be clear this result was indirectly shown by measuring synaptic inhibition in the Ferster and Miller paper. The current study elegantly and rigorously used direct optogenetic stimulation of I cells in transgenic mice to show that 'paradoxical' prediction is borne out in mouse V1. The authors then go on to repeat this core result in S1 and M1 and in deeper cortical layers of these areas. Overall, I find the study well performed and the arguments sound. I previously already subscribed to this view based on the existing data, but the data in this study really is the nail in the coffin. For this reason, this study is timely and warrants publication, with eLife a suitable journal.

Thank you for the positive comments about the experiments and the arguments in the paper.

On existing data, we agree that support for the ISN regime in cat V1, with sensory stimulation, was demonstrated by Ozeki et al. However, our work goes beyond this and other existing results in several ways. Our most important new finding that goes beyond Ferster and Miller is new evidence for ISN operation at baseline (at rest) – without the additional input that comes with sensory stimulation.

Theoretical work on the stabilized supralinear network (SSN; reviewed by Ahmadian and Miller, 2019), shows that networks can be in the non-ISN state at baseline, and then with sensory drive, transition into the ISN state. You showed relevant experimental data in your 2017 Neuron paper, giving evidence for increasing inhibition stabilization (via increasing I/E ratio) as sensory stimulation strengthens. But before our work, an open question had been at what level of activity the ISN transition point occurred – whether it was above or below the level of spontaneous activity, and therefore whether weak inputs were processed by a network operating in the ISN regime or not. Ferster and Miller noted this as well (Ozeki et al., 2009, Discussion section): “Our data [with visual stimulation] do not address whether the network operates as an ISN at rest.”

Our data does show the network operates as an ISN at rest. And our experiments can also give insight into how far spontaneous activity is from the transition. We found that the transition out of the ISN state is substantially below the level of spontaneous activity (as shown in Figure 2L and similar plots.)

We have significantly revised the Introduction to better explain the above topic, including adding more discussion of the non-ISN to ISN transition predicted by Miller and colleagues (Introduction). We appreciate the comments and the opportunity to improve our discussion of these issues.

I do have some questions for the authors to address (both conceptual and methodological). While I agree with almost all of the authors conclusions, I would like them to address a few points in which I will act as a devil's advocate. I'd like them to conceptually, experimentally, or theoretically rule out alternative explanations for the data, however less likely they might seem than the ISN model.1) Could it be that a very small subset of PV+ interneurons are extremely photo-excitable (perhaps a subtype that suppresses all other cell types across all layers in cortex) and that these cells are in fact responsible for the suppression of most other I cells? These cells might be so rare (inhibitory 'hub' cells) that they would be very hard to catch with extracellular recording (or they might have small spikes that are below the noise threshold). Perhaps they have a slightly delayed response to optogenetic stimulation which explains the transient increase in I cell activity prior to its suppression?

Thanks for bringing up these possible alternative explanations.

Our data argues against the possibility of rare, non-paradoxical PV+ interneurons that increase their rate to suppress all other inhibitory cells. We observed that inhibitory cells respond in a non-paradoxical way for large laser intensities, which is inconsistent with the rare-I-cell scenario. If a small number of I cells control the effect, each I cell would be predicted to be either suppressed or excited by the laser, not to transition in the way we observed.

While this firing rate transition (predicted by the ISN model) is the most clear evidence, several other observations also argue against rare inhibitory cells providing feedforward inhibition. First, we found that almost all I cells show paradoxical effects in transgenic animals; e.g. see initial response slopes in Figure 2JK and new Figure 2—figure supplement 7.) Second, the observed dynamics, with initial increase in firing rate followed by suppression in the same inhibitory cells (Figure 3), places tight constraints on such potential rare I cells. As you mention, for stimulated inhibitory cells to show short-latency excitation followed by longer latency feedforward inhibition would presumably require most inhibitory cells to respond quickly to stimulation, but for the rare inhibitory cells to be, oddly, both very photoexcitable but also slow to respond. (Or would require polysynaptic disinhibition of excitatory cells, which would imply unrealistically short synaptic latencies.) Moreover, the transition (from paradoxical to non-paradoxical response with increasing stimulation) happens at the point at which E cells become suppressed, as predicted by ISN models. While it might be possible to achieve this alignment by careful choice of network parameters, it seems unlikely, especially in light of the fact that our ISN model describes these transition points and simultaneously describes the pharmacology data well.

We have added to the Discussion to make these points.

2) Could a pure feed-forward model explain their results coupled with the possibility that L4 interneurons could be more photo-excitable than L2/3 interneurons, despite being deeper in the brain? In this scenario, the 'paradoxical' I cell suppression in L2/3 could be entirely explained by loss of feedforward excitation from L4. It seems like in the data set some I cells didn't show paradoxical effects (and most of the authors data comes from above L4), so maybe this is the case?

This scenario, which was examined by Moore et al., (2018), is also unlikely in our case, for similar reasons as above including our observation that I cells transition from negative firing rate change (paradoxical) to positive change (non-paradoxical) with increasing stimulation. If the main stimulation effect was to excite L4 inhibitory cells, which suppressed excitatory cells, producing loss in feedforward excitation in layer 2/3, we would not expect a slope transition (as in prev. point above) or an unchanged slope in pharmacology, as we observed. Also, in this scenario, our observed response dynamics are hard to explain. In addition, we did not see evidence in depth recordings for increasing excitability in L4 (Figure 7, Figure 7—figure supplement 1), and only a small fraction of I cells show non-paradoxical effects with the transgenic approach (Figure 2J-K).

In addition to revisions indicated above for your prior comments, we have revised the Introduction on Moore et al.,’s data.

3) In any case, the authors should show much more raw data (rather than just average data) in the main figures. I agree it's the average that matters to test the ISN model, but it would be helpful to the reader to see the distribution across units for the various effects: transient activation, shape of 'paradoxical response' etc. Some of this is in the supplement, but not much.

We have added population data to the paper. We had shown the distribution across units for the transient response in the supplement, and we have now moved into the main figures (Figure 3, panel C) a scatterplot showing the transient size vs. the paradoxical suppression effect size. We have also added a set of raw example cells to Figure 2 (new panel I), and have plotted responses for every V1 inhibitory cell in the supplement (new Figure 2—figure supplement 7). We had previously shown the distribution across all cells of the paradoxical effect (initial slope, e.g. Figure 2JK, 5D,H, 6KL etc.), plotted as raw and normalized responses, and we have chosen to retain these. However, these were clearly not immediately interpretable, and we think combining these with the single cell examples, the transient distributions, and the new supplementary figure improves the presentation.